# Dermatologist-like explainable AI enhances trust and confidence in diagnosing melanoma

Tirtha Chanda [1,98], Katja Hauser [1,98], Sarah Hobelsberger[2,98], Tabea-Clara Bucher [1], Carina Nogueira Garcia [1], Christoph Wies [1,3], Harald Kittler [4], Philipp Tschandl [4], Cristian Navarrete-Dechent[5], Sebastian Podlipnik [6], Emmanouil Chousakos [7], Iva Crnaric[8], Jovana Majstorovic [9], Linda Alhajwan[10], Tanya Foreman[11], Sandra Peternel [12], Sergei Sarap[13], İrem Özdemir [14], Raymond L. Barnhill[15], Mar Llamas-Velasco[16], Gabriela Poch[17], Sören Korsing[18], Wiebke Sondermann [19], Frank Friedrich Gellrich [2], Markus V. Heppt[19], Michael Erdmann [19], Sebastian Haferkamp[20], Konstantin Drexler[20], Matthias Goebeler [21], Bastian Schilling [21], Jochen S. Utikal [22], Kamran Ghoreschi [17], Stefan Fröhling [23], Eva Krieghoff-Henning[1], Reader Study Consortium* & Titus J. Brinker [1]✉

Artificial intelligence (AI) systems have been shown to help dermatologists diagnose melanoma more accurately, however they lack transparency, hindering user acceptance. Explainable AI (XAI) methods can help to increase transparency, yet often lack precise, domain-specific explanations. Moreover, the impact of XAI methods on dermatologists' decisions has not yet been evaluated. Building upon previous research, we introduce an XAI system that provides precise and domain-specific explanations alongside its differential diagnoses of melanomas and nevi. Through a three-phase study, we assess its impact on dermatologists' diagnostic accuracy, diagnostic confidence, and trust in the XAI-support. Our results show strong alignment between XAI and dermatologist explanations. We also show that dermatologists' confidence in their diagnoses, and their trust in the support system significantly increase with XAI compared to conventional AI. This study highlights dermatologists' willingness to adopt such XAI systems, promoting future use in the clinic.

Melanoma is responsible for most skin cancer-related deaths worldwide. Early detection and excision are critical for ensuring that patients achieve the best prognosis[1]. However, early melanomas are difficult to distinguish from other skin tumours. Recent advances in artificial intelligence (AI)-based diagnostic support systems in dermatology[1,2] have allowed dermatologists to diagnose melanoma and nevi more accurately with AI support when shown digitised images of suspicious lesions.

While this is a promising development, the evaluation of AI support in clinical practice is substantially impeded by the fact that DNNs lack transparency in their decision making[3]: the General Data Protection Regulation (GDPR) requires all algorithm-based decisions to be interpretable by the end users[4]. Additionally, as found in a recent survey by Tonekaboni et al.[5] clinicians want an understanding of the subset of characteristics that determines a DNN's output. As the ultimate responsibility for a diagnosis lies with the clinician, informed

A full list of affiliations appears at the end of the paper. *A list of authors and their affiliations appears at the end of the paper. ✉e-mail: titus.brinker@dkfz.de

clinicians will justifiably be cautious of employing DNN-based systems without being able to comprehend their reasoning, as DNNs tend to incorporate all correlated features into their decision-making, including spurious correlations[6,7]. Thus, addressing the intransparency of DNNs will allow researchers to comply with the EU Parliament recommendation that future AI algorithm development should involve continual collaborations between AI developers and clinical end users[8,9].

To address the inadequacies of DNN models a variety of explainable artificial intelligence (XAI) methods, i.e., methods aiming to make the reasoning of AI systems more transparent, have been proposed[10]. The two primary branches of XAI techniques are (1) post hoc algorithms that are designed to retrospectively explain the decisions from a given DNN, such as Grad-CAM[11], LRP[12,13] and others[14–16]. They are typically applicable to classes of DNNs that share the same building blocks, such as convolutional layers, and are thus mostly model agnostic. Post hoc XAI is applicable to a variety of scenarios and levels of access to the model parameters, with some algorithms requiring only access to the model input and output[14]. The other branch of XAI techniques are (2) inherently interpretable algorithms[17–20] that are designed to be intrinsically understandable and can thus address concerns about the faithfulness of the explanations to the model that are raised for post hoc XAI[21]. However, these XAI algorithms are tied tightly into the training and model architecture posing additional constraints to both. Besides the manner in which explanatory algorithms are tied into a DNN, XAI algorithms differ in whether they provide explanations globally, i.e., on dataset level[13,16] or class level[18,22], or locally, i.e., for each image individually[11,12,14–16,23]. Furthermore, XAI methods differ in their intended explainee groups: Ribeira and Lapedriza[24] distinguish between AI developers, domain experts and lay people affected by AI decisions, such as patients, who all have different requirements towards a machine explanation and a different level of understanding of the target domain. A diagnosis assistance system requires local, strongly end-user-focussed explanations as doctors need to assess the quality of the machine suggestions on a case-by-case level. A 2022 systematic review on XAI in skin cancer recognition[25] found that out of 29 studies investigating XAI on dermoscopic images only two[17,26] used inherently interpretable XAI methods. The remaining 27 used post hoc algorithms, with content-based image retrieval (CBIR)[15,27] ($n = 7$), CAM[16] ($n = 7$) and Grad-CAM[11] ($n = 5$) being the most common. We assume that inherently interpretable XAI methods have been found to be underrepresented in this review as they require human-understandable concepts for training[17,26] and often come with an unfavourable performance-interpretability trade-off[28]. On the other hand, post hoc XAI methods do not require expert-annotated data and are readily available in many deep learning software libraries[29], resulting in those methods being widely used. However, they are often rejected as solutions to the problem of transparency in the medical domain due to the risk of introducing confirmation bias when interpreting the explanations[21,30].

Rudin[21] objects to the use of post hoc XAI methods for high-stakes decisions, as they typically require the user to interpret the explanations, that is, to interpret why a highlighted image region is relevant or where exactly a relevant feature is located - we refer to this issue as the interpretability gap of XAI. The interpretability gap raises a subtle but critical issue of confirmation bias[31] and thus decreases the trust of informed users in an XAI system in which this gap has not been closed. For example, if the XAI diagnoses a melanoma based on the lower left part of the lesion, the underlying human expectation is that there is a genuine diagnostically relevant feature in the lower left of the lesion. If the dermatologist finds a relevant feature in this area upon closer inspection, they will assume that the machine decision was based on this feature, although they cannot be sure of it since they only know where the XAI paid attention, but not why. The significance of this issue becomes even more apparent if the XAI-highlighted image region does not contain any genuine diagnostically relevant features, leaving the user clueless as to what the XAI decision was based on. Similarly, if the highlighted region contains not only diagnostically relevant features but also features that are spuriously correlated with the diagnosis, such as rulers or surgical skin marks[6,7], the user also cannot be sure whether the diagnosis was made due to genuine features and thus is reliable. From this point of view, the majority of XAI currently used in skin cancer recognition is unable to close the interpretability gap by means of being post hoc[25]. Both inherently interpretable XAI approaches[17,26] included in a recent review on XAI in skin cancer detection[25] use Concept Activation Vectors[18] or a variant thereof[17]. However, both approaches provide dataset-level analyses of the CAVs their neural networks learned. While this approach is well-suited for quality assurance of a DNN, it is unsuitable for the lesion level explanations necessary for a diagnosis assistance system.

Ghassemi et al.[30] have argued that thorough internal and external validation can be used to address concerns about the reliability of DNN models while eschewing the issues introduced by XAI. Although we agree that thorough validation is important, a solution to the transparency problem that closes the interpretability gap and therefore allows users to build (justifiable) trust in a reliable support system is crucial to improve clinicians' diagnostic performance and thus further improve the patient outcome. Besides this, the only large-scale reader study on XAI in dermoscopy with medical professionals[2] found that their XAI was mostly ignored by the study participants leaving an open knowledge gap.

Two recent dermatological XAI systems aim to close the interpretability gap. Lucieri et al.[23] used the clinically well-established and expert annotated concepts from the PH2[32] and derm7pt[33] datasets to create an XAI that provides lesion-level explanations based on concept vectors. Jalaboi et al.[34] employed a specialised light-weight convolutional neural network architecture that was designed to include localisations into training on clinical images of skin lesions. Additionally, they composed an ontology of clinically established terms to explain why the annotated regions are diagnostically relevant. While both works present promising results, the datasets used in both cases are relatively small. Lucieri et al. employ around 7500 nevus and melanoma images from ISIC19[35–37], PH2 and derm7pt for training and evaluation of the diagnostic capabilities of their model, but use a relatively small dataset of 1023 lesions overall for concept training as only PH2 and derm7pt contain annotated concepts. Likewise, Jalaboi et al. utilise a relatively small data set of 554 expert-annotated clinical images of non-cancerous skin lesions. While these dataset sizes are sufficient for a proof-of-concept, they lack a sufficient amount of rare lesion types, such as Blue Nevi, or special cases, such as lesions on the face or palms and soles that present with localisation-specific diagnostic criteria. Additionally, neither of the two studies evaluated the influence of their frameworks on dermatologists.

We therefore introduce a multimodal XAI (Fig. 1a) that provides explanations that (1) close the interpretation gap by localising well-established dermoscopic features[33,38,39] such that (2) they can be interpreted by dermatologists in the initial diagnosis of melanoma, allowing us to (3) assess how clinicians interact with XAI (Fig. 1b). Specifically, we aimed to develop an XAI that is from its core aligned with dermatologists' perspective on melanoma diagnosis. Extending on prior work on explainability in dermatology[34,40], we developed our XAI to provide dermatologist-like localised explanations (Fig. 2a, b). We achieved this goal by employing a well-established, accessible neural network architecture[41] as a network backbone and a dermatoscopic ontology compiled specifically for this task (Table 1) to encompass human expertise. For the training and evaluation of our multimodal XAI, we created an expert-annotated dataset. Additionally, we conducted a three-phase reader study with 116 international participants (Fig. 1b) to quantify our XAI's influence on clinicians. We find that our XAI achieves good diagnostic performance on the distinction

between melanoma and nevus. Its explanations for the task are well-aligned with those used by human experts. While our XAI does not improve the diagnostic accuracy of the clinicians, it increases their confidence in their own diagnosis and their trust in the support system compared to non-explainable AI support. We show that the increase in trust is correlated with the overlap between human and machine explanations. Finally, we publish our dermatologist-annotated dataset to encourage further research in this area. Thus, we take an important step towards the employment of XAI in the clinic to improve patient outcomes by developing a trustworthy well-performing XAI support system that complies with the GDPR, EU recommendations and clinicians' expectations of AI support systems[4,5,9].

## Results

### Our XAI achieves good diagnostic accuracy

We first acquired ground truth annotations of dermoscopic images of melanoma and nevus from 14 international board-certified dermatologists. We used the ResNet50[41] architecture as the backbone for training our XAI model using the acquired annotations. We then evaluated the diagnostic performance of our XAI compared to a ResNet50 baseline classifier trained without annotations. The balanced accuracy of our XAI on the test set was 81% (95% CI: [75.6, 86.3%]) while the baseline classifier achieved 80% (95% CI: [74.4%, 85.4%]) Additionally, we assessed the extent to which our model learned to pay attention to the lesion, as the surrounding skin does not contain meaningful information for the differential diagnosis between melanoma and nevus. To this end, we computed the ratio of the mean Grad-CAM[11] attribution value within the lesion to that outside of the lesion. Our XAI focused on regions inside the lesion significantly more than the baseline classifier ($P < 0.0001$, two-sided Wilcoxon signed-rank test, $n = 196$ images), with ratios of 35.9 (95% CI: [30.7, 42]) versus 4.1, respectively (95% CI: [3.4, 4.7]) (Fig. 3a). Robustness in the presence of artefacts is illustrated in Supplementary Fig. 1.

To further validate the performance of our classifier, we conducted experiments by replacing the backbone network of our XAI and the baseline with eight CNN architectures including ResNet-18, −34,

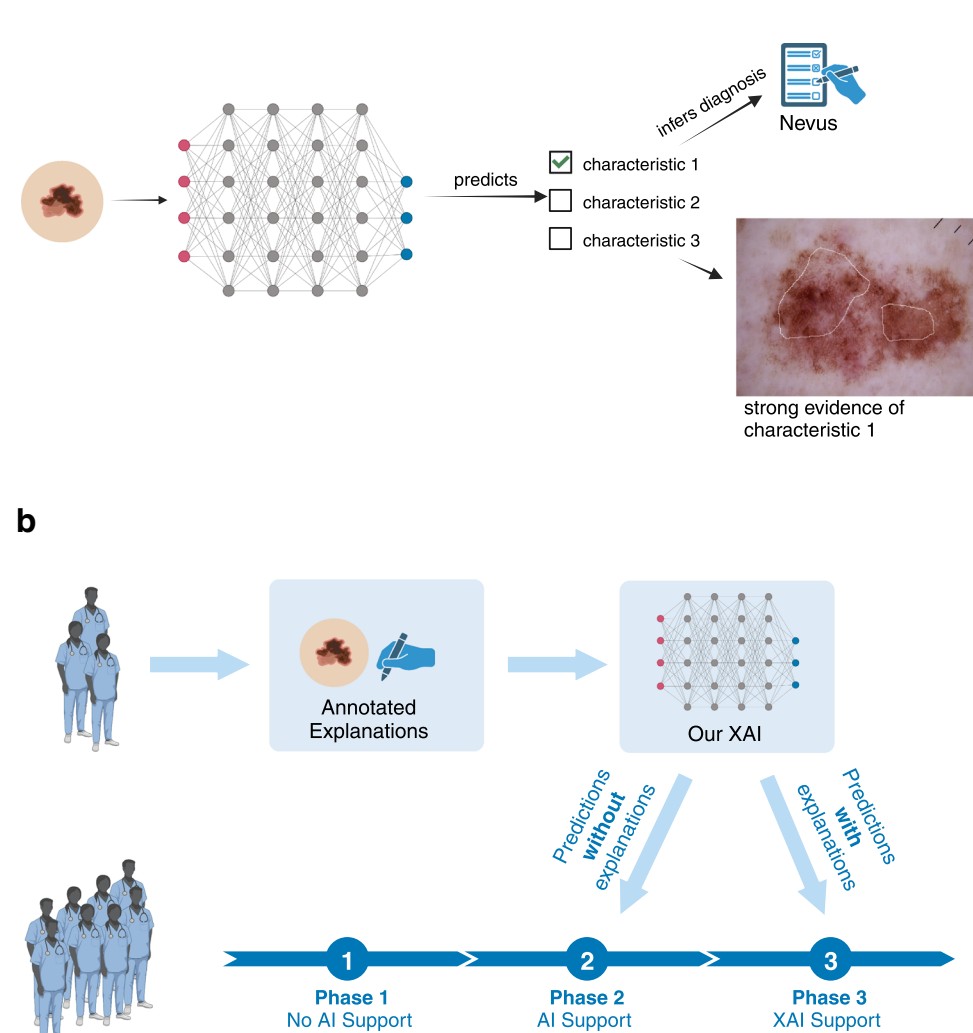

**Fig. 1 | Overview of the XAI and reader study. a** Schematic overview of our multimodal XAI. The AI system makes a prediction for each characteristic and then infers a melanoma diagnosis if it detects at least two melanoma characteristics. The diagnosis and corresponding explanations are then displayed to the clinician. **b** Schematic overview of our work. We first collected ground-truth annotations and corresponding ontology-based explanations for 3611 dermoscopic images from 14 international board-certified dermatologists and trained an explanatory AI on this dataset (top row). We then employed this classifier in a three-phase study (bottom row) involving 116 clinicians tasked with diagnosing dermoscopic images of melanomas and nevi. In phase 1 of the study, the clinicians received no AI assistance. In phase 2, they received the XAI's predicted diagnoses but not its explanations. In phase 3, they received the predicted diagnoses along with the explanations. Figures created with BioRender.com.

**a**

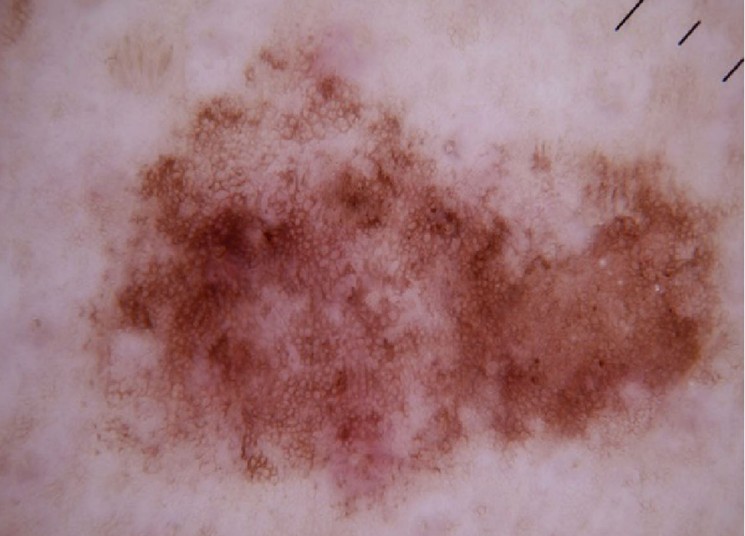

The AI identified this lesion as a **melanoma** with the following characteristics:

strong evidence of
- grey patterns
- thick reticular or branched lines

some evidence of:
- black dots or globules in the periphery of the lesion

**b**

Grey Patterns (strong evidence)

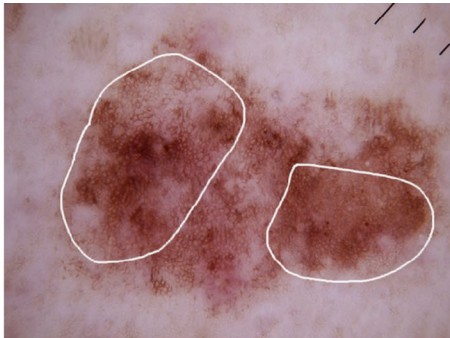

Thick Reticular or Branched Lines (strong evidence)

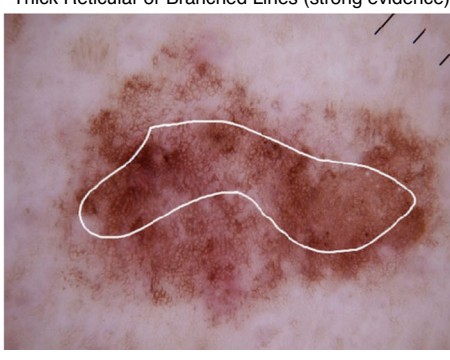

**Fig. 2 | Example multimodal XAI explanation.** An example multimodal explanation from our XAI used in phase 3, showing a textual explanation (**a**) and the corresponding localised visual explanations (**b**). The XAI identified this lesion as a melanoma with the characteristics stated in the textual explanation. The white polygons represent the most important regions where the XAI detected the corresponding characteristics.

−50, −101[41], DenseNet-121, −161[42], EfficientNet-B1, -B3[43]. Our XAI demonstrated superior performance against the baseline on six of the eight architectures, with ResNet-50 obtaining the highest balanced accuracy (Supplementary Table 1). We also compared the performance of our XAI with two additional state-of-the-art approaches. The first approach utilises attention mechanisms and has reported competitive performance on the ISIC 2017 dataset[37,44], while the second approach employs an ensemble of CNN backbones and has achieved the winning position in the ISIC 2020 skin cancer classification challenge[45]. The balanced accuracy of the attention-based approach on the test set was 79% (95% CI: [73.2%, 84.5%]) and the ensemble approach achieved 81.5% (95% CI: [76%, 86.7%]).

To quantify the level of transparency in our XAI and the baseline classifier, we employed a methodology that involves measuring explanation faithfulness. This is achieved through the use of contrastive examples[34,46]. After obtaining the Grad-CAM heatmaps for each image, we randomised all pixels indicated as important for the predictions to create contrastive images. In essence, we obscured the regions that were determined to be significant for the prediction. Next, we computed the difference between the output scores of the original images and that of the contrastive images, and used this as a measure of explanation faithfulness. The faithfulness scores of our XAI and the baseline are shown in Fig. 3b.

Thus, our XAI provides additional interpretability without compromising diagnostic accuracy compared to other state-of-the-art approaches, while also maintaining explanation faithfulness.

## Our XAI is strongly aligned with clinicians' explanations

In phase 1 of the study, participating clinicians were asked to select explanations from an explanatory ontology and localise them on the lesion during diagnosis. We used these explanations to determine the extent to which the XAI system and clinicians detected similar explanations on the same lesions. We evaluated our XAI's alignment with the clinicians' ontological explanations and annotated regions of interest (ROI) by assessing their overlap.

To assess the overlap in ontological explanations, we calculated the Sørensen-Dice similarity coefficients (DSC)[47] between the ontological characteristics selected by the clinicians in phase 1 and those predicted by our XAI. The Sørensen-Dice similarity ranges from 0 to 1, where 0 indicates no overlap and 1 indicates full overlap. The mean explanation overlap was 0.46 (95% CI: [0.44, 0.48], 366 images) when both the clinicians and the XAI predicted melanoma and 0.23 (95% CI [0.20, 0.26], 505 images) when they both predicted a nevus. Considering both diagnoses, the mean explanation overlap was 0.27 (95% CI: [0.25, 0.29], 1089 images) (Fig. 3c). For comparison, we assessed between-clinician overlap to determine the level of agreement among clinicians for the same images. We computed the DSC for each pair of clinician-selected ontological characteristics per image, as each image was diagnosed by multiple clinicians. The mean between-clinician overlap was 0.28 (95% CI: [0.27, 0.29], 5165 pairs) (Fig. 3c), which is comparable to the overlap between the XAI and clinicians.

Next, we investigated the overlap between human and machine ROIs. We defined human ROIs as the image regions that the clinicians annotated to explain their diagnoses in phase 1. For the machine ROIs, we computed the gradient-weighted class activation maps (Grad-CAMs[11]) for the same images, i.e., the image regions that had the biggest influence on the machine's diagnosis. We defined the ROI overlap for an image as the DSC between both sets of ROIs. The mean ROI overlap attained by our XAI was 0.48 (95% CI: [0.46, 0.5]) (Fig. 3c). For comparison, we performed the same calculations using the baseline

**Table 1 | Melanoma and nevus criteria used in our ontology**

| Melanoma criteria | Nevus criteria |
|---|---|
| • Thick reticular or branched lines<br>• Eccentrically located structureless area (any colour except skin colour, white and grey)<br>• Grey patterns<br>• Polymorphous vessels<br>• Pseudopods or radial lines at the lesion margin that do not occupy the entire lesional circumference<br>• Black dots or globules in the periphery of the lesion<br>• White lines or white structureless area<br>• Parallel lines on ridges (acral lesions only)<br>• Pigmentation extends beyond the area of the scar (only after excision)<br>• Pigmentation invades the openings of hair follicles (facial lesions) | • Only one pattern and only one colour<br>• Symmetrical combination of patterns and/or colours<br>• Monomorphic vascular pattern<br>• Pseudopods or radial lines at the lesional margin involving the entire lesional circumference<br>• Parallel lines in the furrows (acral lesions only)<br>• Pigmentation does not extend beyond the area of the scar (only after excision)<br>• Asymmetric combination of multiple patterns and/or colours in the absence of other melanoma criteria<br>• Melanoma simulator |

With the exception of the feature melanoma simulator, the nevus criteria always imply the absence of distinct melanoma features.

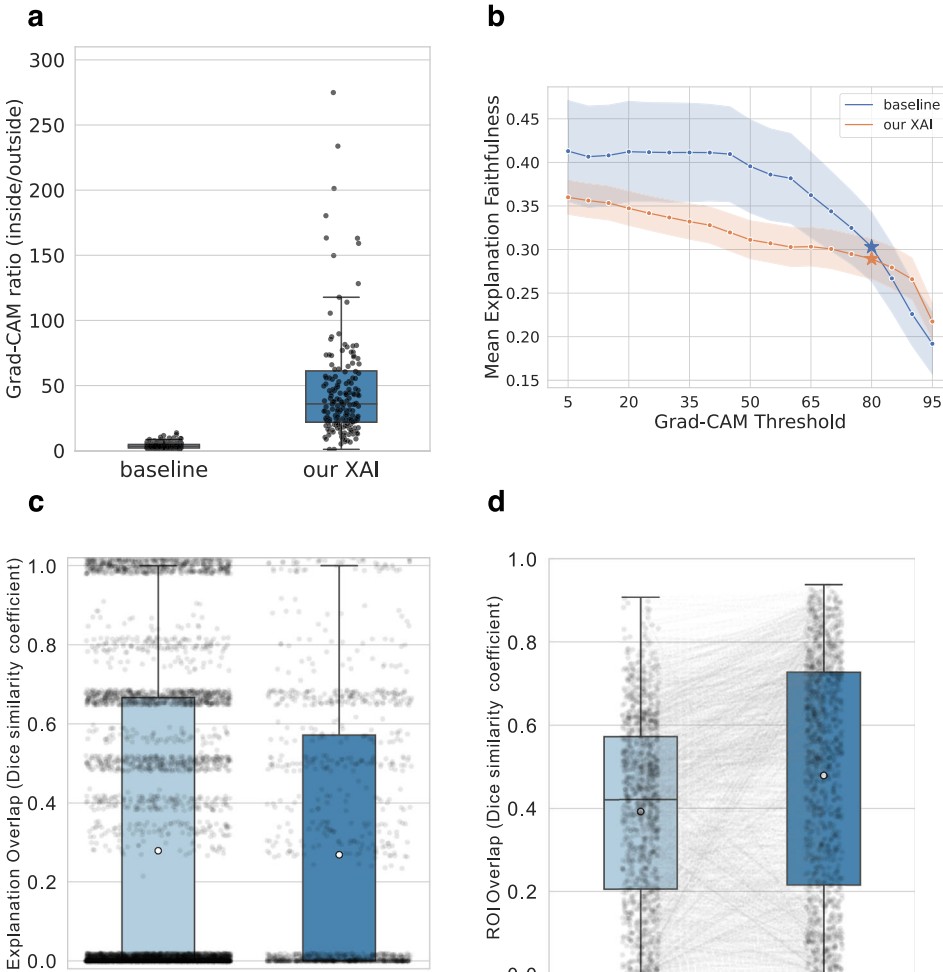

**Fig. 3 | Overview of our XAI's performance. a** Ratio of mean Grad-CAM pixel activation value inside the lesion to that outside the lesion ($P < 0.0001$, two-sided Wilcoxon signed-rank test, $n = 196$ images). Higher values are better, as they indicate greater attention on regions within the lesion than on regions outside the lesion. Four data points for the baseline and 19 data points for the XAI have values above 300 and have been omitted to more clearly visualise the data. **b** We calculated the difference in output scores before and after obscuring the important pixels of the images ($n = 200$ images per threshold). Since we used a threshold on the Grad-CAM heatmaps, we calculated faithfulness values for each threshold ranging from 5 to 95. The stars represent the threshold used in our study and the values of faithfulness at this threshold. The transparent bands represent the 95% bootstrap confidence intervals. **c** Overlap in ontological explanations between clinician pairs for the same image compared to the overlap in ontological explanations between clinicians and our XAI. The whiskers are positioned close to zero and one, and the median lines are positioned close to zero, making them unnoticeable. Each value is shifted by a random number between −0.02 and 0.02 on the y-axis so that the points can be seen more clearly. The between-clinician category consists of $n = 5165$ clinician-pairs, whereas the clinician-XAI category comprises $n = 1089$ images. **d** Region of interest (ROI) overlap between clinicians and our XAI compared to that of the baseline ($P < 0.0001$, two-sided paired t test, $n = 1120$ images). For all boxplots, the horizontal line on each box denotes the median value and the white dot denotes the mean. The upper and lower box limits denote the 1st and 3rd quartiles, respectively, and the whiskers extend from the box to 1.5 times the interquartile range. Source data are provided as a Source Data file.

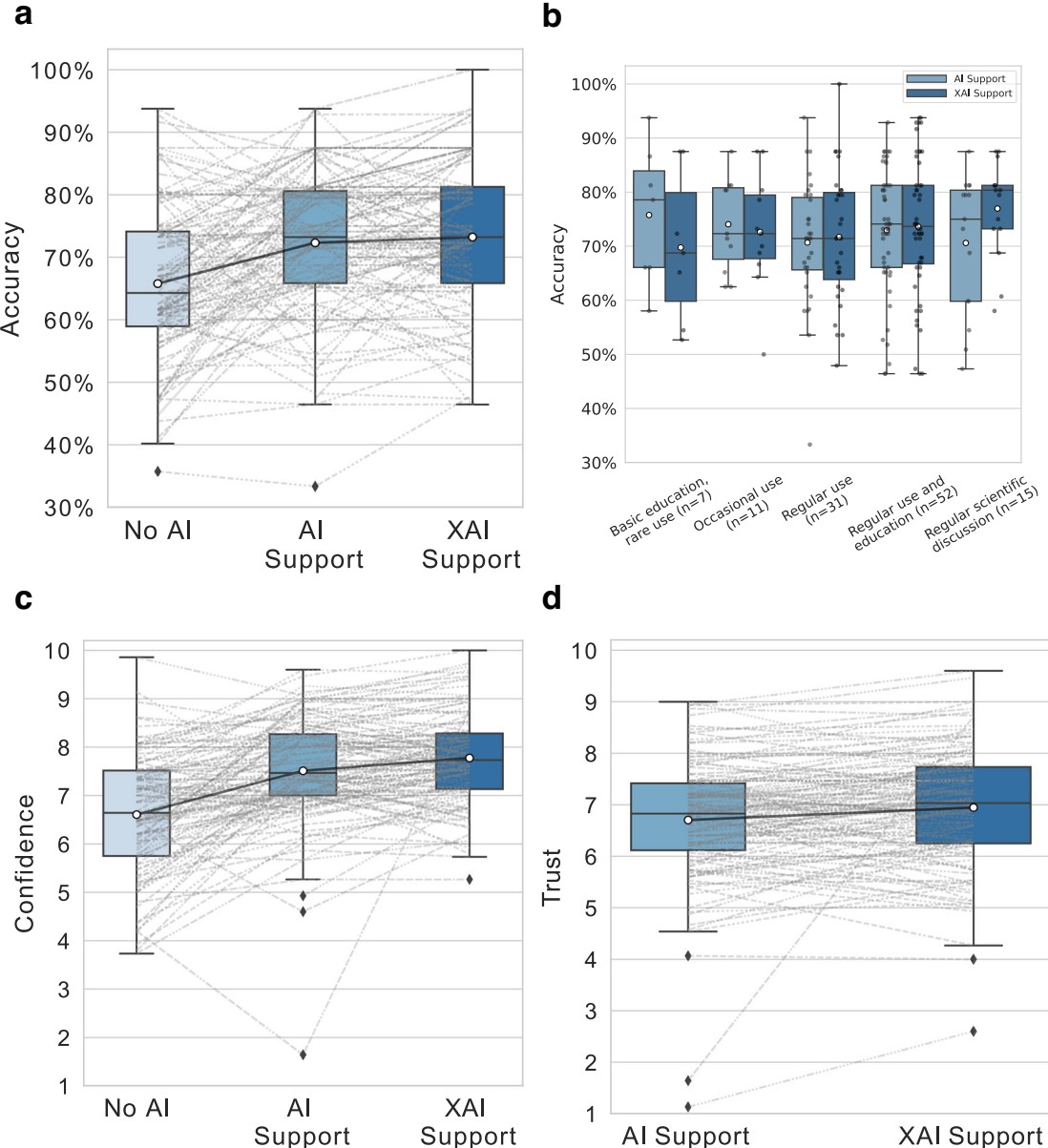

**Fig. 4 | Impact of our XAI on clinicians' diagnostic accuracy, confidence, and trust. a** Distributions of clinicians' balanced accuracy in each phase of our study. ($P < 0.0001$, two-sided paired t test, $n = 109$ participants (No AI vs. AI Support)), ($P = 0.34$, two-sided paired t test, $n = 116$ participants (AI Support vs XAI Support)) **b** Balanced diagnostic accuracy with AI and XAI support grouped by different levels of experience with dermoscopy ($n = 116$ participants). Distributions of clinicians' mean diagnostic confidence ($n = 116$ participants) (**c**) and mean trust in

the support system ($n = 116$ participants) (**d**) in each phase of our study. In figures **a**, **c**, **d**, the grey lines between the phases connect the same participant between phases, and the black lines connecting the boxes indicate the means across all participants. For all figures, the horizontal line on each box denotes the median value and the white dot denotes the mean. The upper and lower box limits denote the 1st and 3rd quartiles, respectively, and the whiskers extend from the box to 1.5 times the interquartile range. Source data are provided as a Source Data file.

classifier and compared them to the overlap of our XAI. The baseline classifier achieved a mean overlap of 0.39 (95% CI: [0.38, 0.41]). Thus, our XAI yielded significantly higher ROI overlap with clinicians than the baseline ($P < 0.0001$, two-sided paired t test, $n = 1120$ images) (Fig. 3d), and we observed similar results in six of the eight architectures (Supplementary Fig. 2). Examples of ROI overlap comparisons are provided in Supplementary Fig. 3.

Thus, the XAI system shows strong alignment with clinicians on both the ontological explanations and the ROI modalities.

**XAI does not further increase diagnostic accuracy over AI alone**
We assessed our XAI's influence on the clinician's diagnostic accuracy compared to both plain AI support and no AI support. To investigate

the relationship between the clinicians' experience and benefit with XAI over AI support, we correlated their change in accuracy with their reported experience in dermoscopy. To differentiate the effect of XAI from the effect of receiving AI support, we first investigated the influence of AI support (phase 2) on the clinicians' diagnostic accuracy against not receiving any AI support (phase 1). Out of 109 participants, we observed a performance improvement with AI support for 77 participants, a decrease for 31, and no change for 1. The clinicians' mean balanced accuracy was 66.2% (95% CI: [63.8%, 68.7%]) in phase 1 and 72.3% (95% CI: [70.2%, 74.3%]) in phase 2 (Fig. 4a). Pairwise comparison revealed a statistically significant improvement ($P < 0.0001$, two-sided paired t test, $n = 109$ participants) by AI support alone.

Next, we investigated whether XAI support (phase 3) had an effect on the clinicians' diagnostic accuracy beyond the benefits of receiving AI support. We compared the clinicians' balanced accuracies between phases 2 and 3 and observed a performance improvement with XAI support for 52 participants, a decrease for 34, and no change for 30. The participants' mean balanced accuracy was 73.2% (95% CI: [71%, 75.3%]) with XAI (Fig. 4a), a slight increase from phase 2. However, a pairwise test revealed that the difference was not significant ($P = 0.34$, two-sided paired t test, $n = 116$ participants). The full details are provided in Supplementary Table 2.

We observed a slight correlation (correlation coefficient 0.2, 95% CI: [0.02, 0.37], $P = 0.03$, Spearman's rank correlation, $n = 116$ participants) between the clinicians' reported experience in dermoscopy and their increase in accuracy with XAI over AI. The clinicians that reported involvement in regular scientific discussions about dermoscopy gained more from XAI support than from AI alone. The group that showed the highest accuracy with AI support alone and also showed the largest decline in accuracy with the use of the XAI reported that they rarely performed dermoscopy (Fig. 4b).

Further analysis revealed that the clinicians were slightly more likely to agree with the machine decisions made in phase 3 than with those made in phase 2. We determined the clinicians' agreement with the AI/XAI by calculating the proportion of images where the human diagnoses matched the machine's predicted diagnoses. The mean agreement was 79.5% (95% CI: [77.1%, 81.2%]) with the XAI in phase 3 versus 77.1% (95% CI: [75%, 79.2%]) agreement with the AI in phase 2. Pairwise testing showed that the mean percentage point increase of 2.4% (95% CI: [0.65% 4.2%], $P = 0.009$, two-sided paired t test, $n = 116$ participants) was significant. We also computed the clinicians' agreement specifically for cases in which the support system was wrong. We observed that the clinicians' agreement on erroneous predictions also increased from 63% (95% CI: [57%, 69.1%]) in phase 2 to 67.9% (95% CI: [61.9%, 73.7%]) in phase 3. However, the pairwise mean increase of 4.8 (95% CI: [−1.2, 10.9]) percentage points was not statistically significant ($P = 0.126$, two-sided paired t test, $n = 116$ participants).

Thus, AI support increased clinician diagnostic accuracy but XAI support did not significantly improve it further, despite the higher agreement. Clinicians with the most experience in dermoscopy benefited most from XAI support, whereas clinicians with less experience benefited most from plain AI support.

## XAI increases clinicians' confidence in their own diagnoses
To compare the influence of AI and XAI support on the confidence clinicians had in their own diagnosis, we compared the participants' confidence scores for each image between phases 2 and 3 for each image. We observed a mean increase of 12.25% (95% CI: [9.06%, 15.74%]) in confidence with XAI support relative to only AI support ($P < 0.0001$, two-sided paired t test, $n = 1714$ images). The mean confidence per participant in each phase is illustrated in Fig. 4c and provided in detail in Supplementary Table 3. The absolute confidence values for the images are reported in Supplementary Fig. 4.

Next, we analysed the influence of displaying the confidence of the XAI on the clinicians' own diagnostic confidence. In phase 3, we observed a slight difference in the clinicians' confidence when the classifier was confident versus when it was uncertain. The mean human confidence score for high-confidence AI predictions was 7.82 (95% CI: [7.73, 7.91]), and that for uncertain AI predictions was 7.69 (95% CI: [7.56, 7.81]) ($P = 0.039$, two-sided Mann–Whitney U test, $n_{high} = 1218$ images, $n_{low} = 496$ images). In phase 2, when the participants received no information about the classifier's certainty, this disparity in the clinicians' confidence was statistically insignificant, with mean confidence scores of 7.54 (95% CI: [7.44, 7.64]) for high-confidence AI predictions and 7.47 (95% CI: [7.31, 7.61]) for low-confidence AI predictions ($P = 0.319$, two-sided Mann–Whitney U test, $n_{high} = 1218$ images, $n_{low} = 496$ images).

In conclusion, the clinicians' confidence in their diagnoses significantly increased with XAI support relative to plain AI support. The XAI's communicated confidence was also correlated with the clinicians' own confidence in their decisions.

## XAI increases clinicians' trust in machine decisions
To assess the impact of the XAI's explanations on the clinicians' trust in the AI's decisions, we compared the trust scores between phase 2 and phase 3 for each image. The mean increase in trust with XAI support in phase 3 was 17.52% (95% CI: [13.74%, 21.6%]), and a pairwise comparison revealed a statistically significant increase relative to only AI support in phase 2 ($P < 0.0001$, two-sided paired t test, $n = 1714$ images). We also observed that trust scores in both phases were significantly dependent on whether or not the clinicians agreed with the AI diagnoses [7.55 (95% CI: [7.48, 7.62]) when they agreed vs. 4.8 (95% CI: [4.64, 4.96]) when they disagreed, $P < 0.0001$, two-sided unpaired t test, $n_{agreed} = 2684$ images, $n_{disagreed} = 744$ images]. The mean trust per participant in each phase is illustrated in Fig. 4d and provided in detail in Supplementary Table 3 and the absolute trust values are given in Supplementary Fig. 4.

Next, we analysed the influence of displaying the confidence of the XAI on the clinicians' trust in the machine predictions. In phase 3, the classifier's transparency about its low certainty did not affect the participants' trust scores. The mean trust score was 6.95 (95% CI: [6.77, 7.14]) for high-confidence AI predictions and 6.96 (95% CI: [6.77, 7.14]) for low-confidence AI predictions ($P = 0.6$, two-sided Mann–Whitney U test, $n_{high} = 1218$ images, $n_{low} = 496$ images). However, we found the same effect in phase 2, where the participants received no AI confidence information: the mean trust score was 6.74 (95% CI: [6.6, 6.87]) on high-confidence AI predictions and 6.67 (95% CI: [6.47, 6.86]) on low-confidence AI predictions ($P = 0.42$, two-sided Mann–Whitney U test, $n_{high} = 1218$ images, $n_{low} = 496$ images).

Hence, clinicians' trust in the machine decisions significantly increased with XAI support relative to plain AI support. Also, the XAI's communicated confidence did not affect the clinicians' trust in the XAI's predictions.

## Clinicians' trust in XAI is correlated with explanation overlap
We hypothesised that clinicians' trust in an AI system is correlated with the amount of overlap between their ontological explanations and the machine's ontological explanations. To determine this, we investigated the correlation between clinicians' trust in AI and the overlap in ontological explanations between the clinicians and the AI. We again defined overlap in ontological explanations as the Sørensen-Dice similarity between the clinician-selected ontological characteristics determined in phase 1 and the XAI-predicted characteristics. To isolate the influence of overlapping explanations on the clinicians' trust in the AI diagnoses, we calculated the overlap and trust correlations by considering only the images where the clinicians' diagnoses matched the AI's diagnoses.

When the clinicians and AI agreed, we observed a slight correlation between trust and overlap in reasoning (correlation coefficient 0.087, 95% CI: [0.02, 0.15], P = 0.01, Spearman's rank correlation, $n = 871$ images) (Fig. 5a). As a sanity check, we assessed the correlation again using the phase 2 trust scores instead. Since no explanations were shown in phase 2, we expected that there would be no discernible correlation between explanation overlap and trust scores. This was indeed the case, as the correlation coefficient was −0.05 (95% CI: [−0.1, 0.01], $P = 0.097$, Spearman's rank correlation, $n = 866$ images).

Upon further investigation, we noticed a difference in the distribution of overlap between the two diagnoses. When both the clinicians and the AI predicted melanoma, the correlation coefficient between trust and the overlap in reasoning was 0.23 (95% CI: [0.19, 0.34], $P < 0.0001$, Spearman's rank correlation, $n = 567$ images), and when both predicted nevus, the correlation coefficient was −0.1

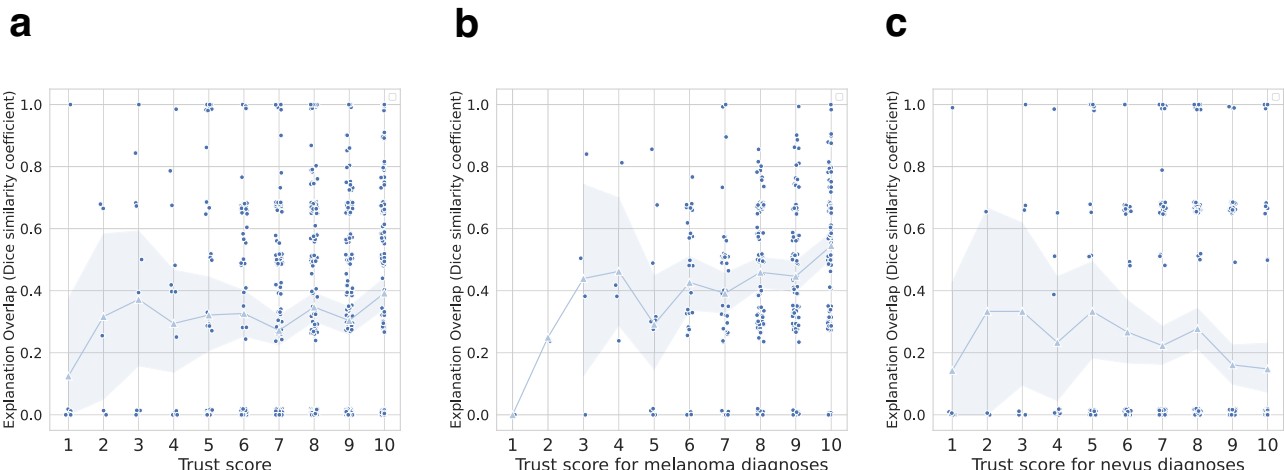

**Fig. 5 | Relationship between clinicians' trust in AI and overlap in ontological explanations. a–c** Correlation between overlap in reasoning (measured by Sørensen-Dice similarity coefficient) and trust in XAI for cases where the clinicians' diagnoses matched those of the XAI ($P = 0.01$, Spearman's rank correlation, $n = 871$ images). The left column depicts the relationship between overlap in reasoning and trust in XAI for both classes (**a**), the middle column depicts cases where both the clinicians and the XAI diagnosed melanoma ($P < 0.0001$, Spearman's rank correlation, $n = 567$ images) (**b**), and the right column represents cases where they both diagnosed nevus ($P = 0.01$, Spearman's rank correlation, n = 505 images) (**c**). Trust is measured on a Likert scale (1–10, with 1 meaning no trust and 10 meaning complete trust in the AI). Each data point is shifted by a random number between −0.02 and 0.02 on the y-axis and −0.1 and 0.1 on the x-axis so that the points can be seen more clearly. The light-coloured triangles connected by lines represent the means (calculated on non-shifted values) of each trust value and the transparent bands represent the 95% bootstrap confidence intervals. Source data are provided as a Source Data file.

(95% CI: [−0.19, −0.02], $P = 0.01$, Spearman's rank correlation, $n = 505$ images) (Fig. 5b, c).

Thus, we found that clinicians place higher trust in machine decisions when their reasoning overlaps. However, this effect was only observable when they diagnosed melanoma compared to when they diagnosed nevus.

## Discussion

Our work intends to close the interpretability gap in AI-based decision support systems by developing an XAI that can produce domain-specific interpretable explanations to aid in melanoma diagnosis. Additionally, we aimed to evaluate the XAI system's effect on clinicians' diagnostic accuracy, confidence, and trust in the system, and to assess the factors that contribute to trust. To this end, we designed a multi-modal XAI system with clinical explanations and conducted the first large-scale reader study on such an XAI system in dermatology.

In our work, we showed that the diagnostic performance of our XAI was on par with the baseline as well as two other state-of-the-art approaches, while being interpretable by learning human-relevant features. To ensure that our XAI's predictive performance was representative, we conducted an evaluation using eight different CNN backbones. We observed that our XAI resulted in improved performance in six out of the eight cases. Additionally, we demonstrated that our XAI was minimally affected by spurious correlations in two ways. First, we quantitatively showed that our XAI was well aligned with clinician-relevant ROIs. Aligning classifier ROIs with clinician ROIs provides a first reassurance that the AI is not making predictions based on spurious correlations. Second, common artefacts that can alter the output score of DNNs used for dermatology[7,48] are almost always located in the area surrounding the lesion and rarely within it. Therefore, we assessed the average pixel attributions within the lesions versus those around them and found that this ratio was significantly greater for our XAI than for the baseline classifier, i.e., we found that the baseline classifier learned false associations significantly more often than our XAI. Hence, we refute the widely accepted notion of performance-interpretability trade-off, and consequently, future AI development can emphasise learning human-relevant features.

We found that AI support improved clinicians' diagnostic accuracy over no AI support, but XAI did not further increase clinicians' diagnostic accuracy compared to AI support alone. The increase in accuracy with AI support is consistent with other studies[1,2]. Therefore, AI assistance can considerably improve diagnostic accuracy, but the effects of XAI still need to be fully explored. Since we observed that experienced clinicians benefit most from XAI and inexperienced ones from plain AI, future research should investigate how experienced versus inexperienced clinicians interact with an XAI as well as the effects of erroneous AI predictions.

We observed that clinicians' confidence significantly increased with AI support and that XAI support enhanced this effect even further. A previous study found that clinicians' confidence did not increase with AI support[1], while another study found an increase in 11.8% of cases[49]. The non-increase in the first study could be attributed to the fact that the participants were shown uncalibrated confidence scores of the AI, which is known to be overconfident even when making incorrect predictions[50]. Seeing the AI exhibit strong confidence in the majority of cases where it may have been incorrect or ambiguous likely prevented an increase in their diagnostic confidence. In contrast, we presented calibrated confidence scores to the participants, which may have had a more favourable effect on their diagnostic confidence. As a result, XAI can increase the adoption of AI assistants since it enhances clinicians' confidence in their decisions.

We showed that clinicians place more trust in a support system's diagnoses when the system explains the reasoning behind its decisions in a way that is interpretable. This aligns with the recent survey by Tonekaboni et al.[5] where clinicians emphasised the need for interpretability. Additionally, we showed that the clinicians' trust in the XAI was correlated with the overlap between the XAI's and clinicians' reasoning. However, while this correlation was significant for images classified as melanoma, we did not observe the same effect for those classified as nevus. When predicting a nevus, melanoma simulator was the XAI's most common explanation, as one ground-truth annotator selected this explanation for the majority of nevus images. Many of the participants did not concur with this explanation, which may explain the lack of overlap-trust correlation for the nevus images. A possible explanation can be found in the work of Grgić-Hlača et al.[51] who

examined the influence of decision aids on humans in certain general tasks like age estimation from face images. They found that seeing a machine decision assistant make errors similar to human errors influences human advice-taking behaviour and that humans perceive similar machine assistance as more accurate, useful, and predictable. When explaining nevus predictions, the XAI made mistakes that humans wouldn't make, i.e., the melanoma simulator explanation, which may account for the lack of correlation in the nevi predictions. We believe additional research on this topic will be beneficial to fully understand the factors that influence clinicians' trust in AI. Nonetheless, XAI can increase clinicians' trust in AI assistance, and as they are more likely to follow the advice of a system they trust, this can result in increased adoption.

A limitation of our work is that, similar to prior works[52], our system was tested under artificial conditions. Furthermore, in contrast to common post-hoc XAI methods, which are intended to make the AI's inner workings transparent without changing the model itself, we intentionally guide our system to provide human-like explanations. This decision comes with the cost of sacrificing the AI's potential ability to recognize patterns that humans do not observe. Nonetheless, we believe that in safety-critical settings, such as melanoma detection, providing explanations that users are familiar with and that they can consequently verify yields a comparably larger benefit. The tight tailoring to a domain-specific ontology poses another major limitation of our work. First, a descriptive framework from which a suitable ontology can be created has to have been established for the desired task. While this is the case for many scenarios in the form of precise diagnostic guidelines (e.g., Gleason scoring[53,54]), it is not always the case. Secondly, as we have shown, clinicians who are familiar with the descriptive system benefit disproportionately from the explanations it provides. This has to be considered in cases such as the dermoscopic differential diagnosis of melanoma and nevus in which no diagnostic guidelines exist and, consequently, no common descriptive framework has been established. Finally, if several descriptive frameworks exist within a domain, such as metaphoric and descriptive terminology in dermoscopy[39], the choice of one system over the other has the potential to alienate some users. However, these are issues that are best addressed by domain-specific measures, such as standardisation, or close consultation with expert boards when creating an ontology. Another constraint to consider in our study design is that the effect of the multimodal XAI can only be attributed to the combination of the explanation modalities. The additional investigation required to analyse the effect of the individual components alone would have significantly surpassed the study's scope. Nevertheless, we have a clear objective to explore the effects of each unique explanation component in future work.

Additionally, our work does not deal with the domain shift problem, i.e., a situation in which the data encountered after deployment in the clinic is significantly different from the data the AI was trained on. The performance of our classifier must be extensively verified in domain shift scenarios, with the addition of unsupervised domain-adaptation techniques[55], such as Adversarial Residual Transfer Networks[56]. In the future, we plan to expand on our ground-truth dataset and include images from multiple clinics, which may improve the generalizability of our XAI to multiple clinics.

Our work advances the field of human-computer collaboration in medicine by developing a multimodal XAI that generates human-like explanations to support its decisions and evaluating its influence on clinicians. The EU Parliament recommends that future AI algorithm development should involve continual collaborations between AI developers and clinical end users[8,9]. Our XAI design was guided by the end user perspective and can iteratively be improved with the results of this study, further clinician feedback, and future work on XAI-human collaboration. The findings of our reader study illustrate the ability of XAI to enhance clinicians' diagnostic confidence and furthermore

suggest that it has the potential to increase the likelihood of considering machine suggestions. The European General Data Protection Regulation (GDPR) requires that all algorithm-based decisions be interpretable by the end users[4]. As our work addresses the current discrepancy between legal and ethical requirements for transparent clinical XAI systems and state-of-the-art classifiers, it constitutes an important first step towards closing the interpretability gap.

## Methods

### Inclusion and Ethics
Our research complies with all ethics regulations. The study's ethics vote is held by the University Clinic Mannheim of the Medical Faculty of the University of Heidelberg. Informed consent was collected from all participants. We did not collect any data on sex and gender of the clinicians participating in our reader study. As compensation, we offered them the opportunity to be credited as a collaborator of our work, which we have listed in Supplementary Data 1.

### Software and Statistics
Prior to data collection, we registered our hypotheses and analysis plan on the Open Science Framework website (https://osf.io/), which may be accessed under https://osf.io/g3keh. We followed the STARD guidelines[57], which we report in detail in Supplementary Information A. All code was written in Python (3.9.9). PyTorch (1.10.0), PyTorch Lightning (1.5.10), Albumentations (1.0.3), NumPy (1.22.2), Pandas (1.4.0), SciPy (1.8.0), OpenCV (4.5.5), Scikit-learn (1.1.0), Matplotlib (3.1.1), and Seaborn (0.11.2) were used for image processing, model development and training, data analysis, and visualisation. All pairwise significance testing was performed using the two-sided paired t test. We utilised the Mann–Whitney U test to determine the significance of the difference between the trust and confidence scores for high- and low-confidence AI predictions and the Wilcoxon signed-rank test to determine the significance of the difference in mean pixel activation ratios due to the nonnormality of the distributions. To calculate confidence intervals, we utilised the bootstrapping method with 10000 samples and a random seed of 42 each time the confidence interval was calculated. We set an alpha value of 0.05, and the P values were adjusted using the Bonferroni method to correct for multiple comparisons.

### Explanatory ontology
To allow for speedy annotation by both our annotating dermatologists as well as the study participants in phase 1 of the reader study and to facilitate a streamlined evaluation of the explanations, we created an ontology containing typical features of melanomas and nevi based on pattern analysis. We combined well-established features from several sources[33,38,39] and included feedback from the dermatologists participating in the pilot study (see Supplementary Information B for details) as well as from the ground-truth annotators. The ontology was first compiled in German and approved by a panel of board-certified dermatologists prior to the study. The German-speaking participants of the three-phase reader study received the original German version of the ontology. We translated the ontology to English for the international study participants. The translation was approved by two board-certified dermatologists (RB, MLV). All features are listed in Tab. 1. More details on the ontology can be found in Supplementary Information C.

### Images and annotation procedure
We used the publicly available dataset HAM10000[35] for our study, which contains 10015 dermoscopic images of several skin diseases at different localisations on the body. The dataset contains images from both sexes and patient age ranges from 0 to 85 reported in 5-year intervals. During the construction of the dataset, several images per lesion were often taken, and occasionally, more than one lesion per

patient was included. Thus, the number of images is greater than the number of unique lesions, and the number of unique lesions is greater than the number of patients. The diagnoses were confirmed by excision and subsequent pathological evaluation, by panel decision or by follow-up.

In this study, we used all the biopsy-verified melanoma and nevus images in the HAM10000 dataset, i.e., a set of $n = 3611$ images of 1981 unique lesions. We refer to this set of images as the base set in the remainder of this section. To acquire the necessary annotations for training the classifier, we asked 14 international board-certified dermatologists to annotate these 3611 images of biopsy-verified melanomas and nevi from the dataset. To prevent any data loss as a result of misdiagnoses, we provided the ground truth diagnoses to the annotators. With knowledge of the diagnosis of the lesion, the annotating dermatologists were tasked with explaining the given diagnosis by selecting the relevant features from the explanatory ontology and by annotating the image regions of interest (ROIs) corresponding to the selected features. One dermatologist (SHo) annotated all 3611 images, while each of the other 13 annotators annotated between 200 and 300 unique lesions such that our dataset contains annotations by at least two dermatologists per unique lesion. We set the explanatory labels for an image as the union of the annotator's explanatory labels, and we merged the ROIs according to the information loss of the merged ROI relative to the original ROIs (full details can be found in Supplementary Fig. 5 and Supplementary Information D).

We split the base set into a training set, a validation set and a test set. The test set contained 200 unique lesions with 100 unique melanomas and nevi each. For this, we randomly chose 100 unique melanomas and nevi (with complete information on patient age and sex, as well as on the localisation of the lesion) from the base set. For the test set, we kept only one image per lesion; in cases where several images were present for a single lesion, we chose the last image as identified by the image ID. After assigning images to the test set, we proceeded by removing all images that contained the selected lesions as well as other lesions from the same patients from the base set. We then performed a random 82:18 split on the unique lesions in the remainder of the base set to form the training set and the validation set, respectively. In doing so, we ensured that all images of lesions that were photographed multiple times as well as lesions from the same patient were contained in only one of the sets to avoid leaking information from the training to the validation set. As a result, our training set contained 2646 images of 1460 lesions, and the validation set contained 599 images of 321 lesions. Around 22% of the lesions in each set were melanomas, while 78% were nevi.

## XAI Development

**Classifier design.** We developed an AI classifier that is able to explain itself to clinicians by making use of the well-established visual characteristics[33,38,39] of melanoma and nevi from our explanatory ontology. Our classifier learns to predict these characteristics from digitised dermoscopic images and infers the diagnosis of melanoma or nevus from its predictions.

After acquiring the ground-truth annotations from the dermatologists, we trained the classifier on the annotations to predict the lesion characteristics. Utilising the annotations optimises our XAI to be aligned with dermatologists' perspective on melanoma diagnosis. We follow the attention inference architecture introduced by Li et al.[40] and extended by Jalaboi et al.[34] Our classifier has two components: a classification component $Comp_C$ and a guided attention component $Comp_A$ to help localise the relevant features. In $Comp_C$, instead of predicting the diagnosis directly, the classifier predicts the characteristics from our ontology. We infer the diagnosis as melanoma if at least two melanoma characteristics are detected; empirically, we found that this approach leads to the best trade-off between sensitivity and specificity, and clinically, this approach is similar to the use of the 7-point

checklist[38], which also requires at least two melanoma criteria for a diagnosis of melanoma if used with the commonly used threshold of three points.

To guide the classifier to learn features used by dermatologists and create more meaningful explanations, we employ $Comp_A$. For training, we define the loss $L_A$ in addition to the regular cross-entropy loss between the target and the prediction as Eq. 1:

$$L_A = \frac{1}{N} \sum_{i=1}^{N} \left( 1 - \frac{1}{C} \sum_{c=1}^{C} \frac{2A_{i,c}H_{i,c}}{A_{i,c} + H_{i,c}} \right) \quad (1)$$

where $N$ is the number of samples, $C$ is the number of classes, $A_C$ is the Grad-CAM[11] attention map generated from the last convolutional layer of $Comp_A$ and $H_C$ is the ground-truth ROI annotated by the dermatologists. For images where the ground-truth label of a characteristic $c$ was 0, i.e., the characteristic was not present in the lesion, we set $H_C$ to be a zero-valued matrix of the same size as $A_C$. This additional loss term was added to the regular cross entropy loss to yield the following combined loss as defined in Eq. 2:

$$L = \lambda_C L_C + \lambda_A L_A \quad (2)$$

where $L_C$ is the cross entropy loss for the characteristics and $L_A$ is the Dice loss between the Grad-CAM heatmaps of the model's predictions and the ROIs annotated by the dermatologists. $\lambda_C$ and $\lambda_A$ are hyperparameters for assigning weights to the individual components. For all our experiments, we set $\lambda_C$ to 1 and $\lambda_A$ to 10. A graphical illustration of this design can be found in Supplementary Fig. 6.

We opted to use a ResNet50 pretrained on the ImageNet dataset[58] as a feature extractor since it has been shown to perform well in skin lesion classification tasks[59]. After the feature extraction backbone, we added a dropout layer and an output layer of one neuron. We used random sampling to balance the class distribution during training. We also used several image augmentations to improve generalizability, in line with the International Skin Imaging Challenge (ISIC) skin cancer classification challenge winners, who achieved state-of-the-art performance on a skin lesion classification task[45]. Complete details on the model hyperparameters can be found below.

Additionally, we chose to display the confidence of the classifier for each prediction. Conventionally, the raw output of the softmax or sigmoid layer is used as a measure of confidence; however, this value is an unreliable measure of confidence and should be calibrated[50]. To obtain well-calibrated probabilities, we performed temperature scaling on each output class, which is a simple but effective method for calibrating neural network outputs to more accurately reflect model confidence[50].

**Classifier performance testing.** We evaluated the performance of the classifier on the held-out test set in terms of balanced accuracy. The sensitivity and specificity of the classifier were determined on the validation set. For the calculation of the ratio of mean Grad-CAM attributions within the lesion to those surrounding the lesion, we used the formula as defined in Eq. 3:

$$Ratio = \frac{\mu(Grad-CAM\ attributions\ within\ lesion)}{\mu(Grad-CAM\ attributions\ surrounding\ lesion)} \quad (3)$$

We determined the regions inside and outside of the lesions using the HAM10000 segmentation maps[2].

For the calculation of overlap between the XAI-predicted explanations and the clinician-selected explanations, we used the Sørensen-Dice similarity coefficients (DSC), calculated with the numbers of true

positives (*TP*), false positives (*FP*), and false negatives (*FN*) as in Eq. 4:

$$DSC = \frac{2TP}{2TP+FP+FN} \quad (4)$$

We also used the DSC for Regions of Interest (ROI) comparisons, this time calculated as in Eq. 5, where $a$ and $b$ are the soft image masks and $\epsilon$ is a smoothing term:

$$DSC = \frac{2\sum(a \cdot b) + \epsilon}{\sum a + \sum b + \epsilon} \quad (5)$$

**Design of the explanations.** According to the EU Parliament's recommendations regarding AI in healthcare, future AI algorithm development should be based on co-creation, i.e., continual collaborations between AI developers and clinical end users[8,9]. Consequently, we designed our explanation scheme with the consultation of two board-certified dermatologists (SHo, CNG). The explanation scheme included both visual and text-based components as well as assessments of the classification confidence.

The majority of AI explanation approaches are visual, using saliency maps to emphasise the areas in an image that are most important in making predictions. This is most commonly achieved by superimposing a rainbow-coloured heatmap onto the image, but other visualisations are also possible. Heatmap methods were initially created with AI developers' debugging needs in mind, as they allow by means of their colour gradient for a fine-grained analysis of the importance of image regions. However, according to the consulting dermatologists, such heatmaps obscured their view of the lesion so that they needed to switch back and forth between the explanation image and the original image without the heatmap. This was deemed tedious and unsuitable for the diagnostic process. The dermatologists expressed the need for a clear view of the lesion in the explanation image, allowing them to quickly determine whether the predicted features are present in the salient regions. Therefore, we decided to indicate the most relevant region(s) for the prediction of each feature by displaying a polygon-shaped ROI over the top 20th percentile attribution values, as shown in Fig. 2b, based on the guidance of the consulting dermatologists. Only showing a polygon has the drawback that fine-grained localisation information within the polygon is lost and that obtaining ROIs that are neither too precise nor too general becomes dependent on choosing a suitable threshold. Additionally, the person seeing the explanation has to interpret the region within the polygon as the more important region. We experimented with a slight darkening of the unimportant regions outside of the polygon to solve this issue. However, we rejected this option for our study design, as the consulting dermatologists pointed out that it limited their ability to assess the regions outside of the polygons and, in our test set, all salient regions were contained inside the polygons. This was also indicated in the survey for clarity. However, our polygon approach is, as is, unsuitable for application in the clinic, as different behaviour must be anticipated, especially in degenerate cases. Nevertheless, we believe that a medical device using heatmap-based explanations should offer several interactive modes anyway. We imagine such a tool to allow users to switch between heatmap and polygon explanations and to select different levels of opacity or importance thresholds, allowing them to intuitively determine which reasons are important while limiting interference with the visibility of the lesion. Using a similar scheme as presented in Lucieri et al.[23], we also provide a textual explanation of the characteristics detected in a lesion.

Additionally, in clinical applications, it is essential that the AI system be able to communicate when its predictions are uncertain[60,61]. This allows clinicians to judge when they should trust the AI predictions and when to disregard them. In our study, we presented the degree of confidence for the detection of each characteristic.

Predictions of characteristics with high confidence were displayed with the text "strong evidence of characteristic(s)" while those with low confidence were displayed with the text "some evidence of characteristic(s)". We say that the classifier is certain when it finds strong evidence of at least one characteristic (temperature-scaled output above 0.7) and is uncertain otherwise. An example of this is provided in Fig. 2a. This communicates the prediction uncertainty to the dermatologist, as the absence of strong evidence of all characteristics indicates that the classifier is not confident. This explanation scheme was illustrated to the study participants in a tutorial video in phase 3 (https://youtu.be/eWAcaIzXChY). The threshold was set based on prior research showing an improvement in performance when rejecting uncalibrated output scores below 0.7[62]. We note that the prior work worked with uncalibrated output scores, whereas we use output scores that have been calibrated by temperature scaling.

The localised explanations in phase 3 were created by showing the regions of interest for the characteristics the classifier was certain about (respective outputs above 0.7). If the classifier identified no characteristic above the certainty threshold, we showed the regions of interest for the most certain characteristic instead.

**Hyperparameters of the classifiers.** Except for the number of epochs, the hyperparameters for both our XAI and the baseline are identical. The XAI required a greater number of epochs than the baseline because it had a greater number of target classes. For the attention-based baseline and the ensemble baseline, we used the same hyperparameters as in the original works, which are listed below.

**Our XAI.** Hyperparameters: backbone=ResNet50, num_epochs=30, learning_rate=0.0001, optimizer=Adam(epsilon=1e-08), batch_size=32, image_size = (224, 224), dropout=0.4, seed=42.

Image augmentations: Transpose ($p = 0.2$), VerticalFlip ($p = 0.5$), HorizontalFlip ($p = 0.5$), ColorJitter ($p = 0.5$), CLAHE (clip_limit = 4.0, $p = 0.7$), HueSaturationValue (hue_shift_limit = 10, sat_shift_limit = 20, val_shift_limit = 10, $p = 0.5$), ShiftScaleRotate (shift_limit = 0.1, scale_limit = 0.1, rotate_limit = 15, border_mode = 0, $p = 0.85$), Resize (image_size, image_size), Normalize().

**Baseline ResNet50 classifier.** Hyperparameters: backbone = ResNet50, num_epochs = 25, learning_rate = 0.0001, optimizer = Adam(epsilon = 1e-08), batch_size = 32, image_size = (224, 224), dropout=0.4, seed=42.

Image augmentations: Transpose ($p = 0.2$), VerticalFlip ($p = 0.5$), HorizontalFlip ($p = 0.5$), ColorJitter ($p = 0.5$), CLAHE (clip_limit=4.0, $p = 0.7$), HueSaturationValue (hue_shift_limit = 10, sat_shift_limit = 20, val_shift_limit = 10, $p = 0.5$), ShiftScaleRotate (shift_limit = 0.1, scale_limit = 0.1, rotate_limit = 15, border_mode = 0, $p = 0.85$), Resize(image_size, image_size), Normalize().

**Baseline Attention-based classifier.** Hyperparameters: backbone = InceptionResNetV2, num_epochs=150, learning_rate=0.01, optimizer = Adam(epsilon = 0.1), batch_size = 50, image_size = (224, 224), dropout = 0.5, seed = 42.

Image augmentations: rotation_range = 180, width_shift_range = 0.1, height_shift_range = 0.1, zoom_range = 0.1, horizontal_flip = True, vertical_flip = True, fill_mode=nearest.

**Baseline Ensemble classifier.** Hyperparameters: The hyperparameters are listed in Table 2.

Image augmentations: Transpose ($p = 0.5$), VerticalFlip ($p = 0.5$), HorizontalFlip ($p = 0.5$), ColorJitter ($p = 0.5$), OneOf([MotionBlur (blur_limit = 5), MedianBlur (blur_limit = 5), GaussianBlur (blur_limit = (3, 5)), GaussNoise(var_limit = (5.0, 30.0))], $p = 0.7$), OneOf([OpticalDistortion(distort_limit = 1.0), GridDistortion(num_steps = 5, distort_limit = 1.), ElasticTransform (alpha = 3)], $p = 0.7$),

**Table 2 | Hyperparameters of the baseline ensemble classifier**

| backbone | Init_lr | epochs | image_size | dropout |
|---|---|---|---|---|
| EfficientNetB1 | 3e-5 | 18 | 224, 224 | 0.5 |
| EfficientNetB4 | 3e-5 | 15 | 224, 224 | 0.5 |
| EfficientNetB4 | 3e-5 | 15 | 224, 224 | 0.5 |
| EfficientNetB4 | 3e-5 | 15 | 224, 224 | 0.5 |
| EfficientNetB4 | 2e-5 | 15 | 224, 224 | 0.5 |
| EfficientNetB4 | 3e-5 | 15 | 224, 224 | 0.5 |
| EfficientNetB5 | 3e-5 | 15 | 224, 224 | 0.5 |
| EfficientNetB5 | 1.5e-5 | 15 | 224, 224 | 0.5 |
| EfficientNetB5 | 1.5e-5 | 15 | 224, 224 | 0.5 |
| EfficientNetB5 | 3e-5 | 15 | 224, 224 | 0.5 |
| EfficientNetB6 | 3e-5 | 15 | 224, 224 | 0.5 |
| EfficientNetB6 | 3e-5 | 15 | 224, 224 | 0.5 |
| EfficientNetB6 | 3e-5 | 15 | 224, 224 | 0.5 |
| EfficientNetB7 | 3e-5 | 15 | 224, 224 | 0.5 |
| EfficientNetB7 | 1e-5 | 15 | 224, 224 | 0.5 |
| EfficientNetB7 | 1e-5 | 15 | 224, 224 | 0.5 |
| SE_ResNext101 | 3e-5 | 15 | 224, 224 | 0.5 |
| ResNest101 | 2e-5 | 15 | 224, 224 | 0.5 |

CLAHE (clip_limit = 4.0, $p = 0.7$), HueSaturationValue(hue_shift_limit=10, sat_shift_limit = 20, val_shift_limit = 10, $p = 0.5$), ShiftScaleRotate(shift_limit = 0.1, scale_limit = 0.1, rotate_limit = 15, border_mode = 0, $p = 0.85$), Resize (image_size, image_size), CoarseDropout (max_height=int(image_size * 0.375), max_width = int (image_size * 0.375), max_holes = 1, $p = 0.3$), Normalize().

## Study design

Our reader study consisted of three parts and took place between July and December 2022.

**Participants.** We recruited a total of 120 international clinicians specialised in dermatology for phase 1, 116 of whom finished the complete study. The participants were contacted via Email through our collaboration network and by using public contact data from the International Society for Dermoscopy website and from university clinic webpages. We also contacted participants from private clinics.

We excluded the data of participants who entered constant values for trust, confidence, and/or diagnosis, such as entering a trust score of 7 for all 15 images or a diagnosis of nevus for all 15 images. We excluded images where the participant took less than 7 seconds to complete. We did not use an upper limit for time taken for exclusion because some complicated cases could take a long time to annotate. Furthermore, the participants could pause and resume working later, so a longer time taken did not necessarily imply insincere work. Images marked as having insufficient image quality ($n = 26$) were removed for the particular participant who indicated the issue, but not for others since the image quality issues could have been related to monitor settings. As a result, a varying amount of images were evaluated for each participant. None of the participants met these criteria for exclusion. Participants who dropped out in phases 2 or 3 were excluded from the study.

**Phase 1.** Phase 1 of the study took place between July and October 2022. We tasked the clinicians to diagnose 15 lesions from our dataset, to explain their diagnoses by choosing the relevant characteristics from our explanatory ontology and to annotate the characteristics in the given images. Furthermore, we asked the clinicians to indicate their confidence in their diagnosis. The participants were informed that they would be presented with 15 lesions (14 unique and one repeated image) each and that this phase would take up to 30 minutes to complete based on the experience from our pilot study. The

participants were not informed about the repeated image. The participants were asked to complete this task within two weeks.

We randomly divided the participants into 14 groups. Each group contained roughly 4–6 participants. For each group, we randomly selected 14 images (7 melanomas and 7 nevi) from our test set (196 images in total, with 98 melanomas and 98 nevi) and repeated the third image in the group (either a melanoma or a nevus) after the 12th image. The image sets for each group were mutually exclusive and consisted of 196 unique images (see Supplementary Data 2 for the image IDs used in each group). The test set was drawn at random from our dataset and curated to contain only one image per lesion and one lesion per patient. All images from the patients contained in the test set were removed from the training and validation set. We used the repeated image to measure the variability of results for the same participant, but we did not exclude any participants from our analysis based on this variability.

We asked the clinicians to diagnose each lesion as a nevus or melanoma. To reflect the clinical practice of excising lesions that are not considered to be unequivocally benign and the German dermatology guideline to excise specific types of nevi, we offered the diagnoses "nevus (leave in)", "nevus (excise)," and "melanoma". For the evaluation of clinician accuracy, we treated both options for nevus as a simple "nevus" diagnosis.

In addition to the diagnosis, we asked the participants to choose one or more characteristics from the explanatory ontology and to annotate the corresponding image regions (ROIs). Finally, the clinicians were asked to indicate their confidence in their diagnosis on a Likert scale (1–10, with 1 being least and 10 most confident). The participants had the option to indicate issues during the processing of the survey (i.e., "insufficient image quality", "no image visible", "no AI diagnosis visible", and "other", the latter being accompanied by a free text field).

To conduct this part of the study, we used the web-based annotation tool PlainSight (https://plainsight.ai/). The clinicians received textual information on the study as well as a video (https://youtu.be/BJRq4nXZ1Xw) explaining use of the tool, the explanatory ontology, and annotation of the ROIs with their login details.

3 participants dropped out in this phase leaving 113 participants.

**Phase 2.** The second phase of our study was conducted in November 2022. In this phase, we included the 113 participants who completed phase 1 and 7 participants from our pilot study. We asked the participants who completed phase 1 to diagnose the same lesions they reviewed in phase 1 with the support of an AI system but did not explicitly inform them that they had diagnosed the same lesions in the previous phase. The 7 pilot study participants had not previously reviewed these lesions as we used a different set of images. We ensured that at least two weeks had passed between finishing phase 1 and starting phase 2. Again, the participants were asked to complete the task within two weeks.

The participants were shown the images from phase 1 in the same order alongside the AI diagnosis of the lesion ("nevus" or "melanoma") and asked them to provide their own diagnosis. As in phase 1, they could choose between "nevus (leave in)", "nevus (excise)" and "melanoma". The participants were informed of the AI's sensitivity and specificity.

As in phase 1, we asked the participants to indicate their confidence in their decisions on a Likert scale (1–10, with 1 being least and 10 being most confident). Additionally, we asked them to indicate their trust in the AI decision on a Likert scale (1–10, with 1 meaning no trust and 10 meaning complete trust in the AI) in this phase.

They were informed that the assessment would take 10–12 minutes to complete. We used the web-based survey tool LimeSurvey (https://www.limesurvey.org/) to conduct this phase.

3 participants did not complete this phase before the deadline, resulting in 117 participants.

**Phase 3**. The final phase of the study was conducted in December 2022. Again, we ensured that at least two weeks had passed between the completion of phase 2 and the start of phase 3. In line with previous phases, the time given for the task was two weeks.

Of the clinicians who completed phase 2, those who participated in phase 3 ($n = 117$) were asked to diagnose the same lesions as in the previous study phases, this time with the support of an explainable AI. Again, they were not informed that they had diagnosed the same lesions in the previous phase or that an image had been repeated, but similar to phase 2, they were informed of the AI's sensitivity and specificity.

For each feature that was detected with certainty (temperature-scaled softmax output >0.7), we showed a separate explanation. If the AI did not detect any feature with certainty, we showed the explanation for the feature with the highest AI confidence. The participants were informed that they were receiving explanations for "strong evidence for feature(s)" or "weak evidence for feature(s)". The explanations always followed the same schema: the clinicians were shown the relevant entry from the ontology as a textual explanation and the location of the feature based on the highest-influence region(s) of the AI's Grad-CAM saliency map (0.7 or higher). An example is shown in Fig. 2a, b.

Similar to phase 2, the participants were asked to indicate their confidence in their decisions and their trust in the AI decisions. They had the same diagnosis options and the possibility of indicating issues that arose during the assessment. As in phase 2, we used LimeSurvey to conduct this phase and provided the clinicians with a video (https://youtu.be/eWAcaIzXChY) on how to interpret the AI explanations. With 1 participant dropping out, a total of 116 participants completed this phase (82 board-certified and 33 resident dermatologists as well as one nurse consultant specialised in dermoscopic skin cancer screening).

### Reporting summary

Further information on research design is available in the Nature Portfolio Reporting Summary linked to this article.

## Data availability

The dermoscopic images used to train, validate, and test our classifier are publicly available from the HAM10000 dataset (https://doi.org/10.1038/sdata.2018.161) and can be accessed here: https://dataverse.harvard.edu/dataset.xhtml?persistentId=doi:10.7910/DVN/DBW86T. The images used in our work can be filtered by selecting the images of biopsy-verified melanoma and nevi. We used ImageNet weights to pretrain our classifier (https://pytorch.org/vision/stable/models.html). The data generated in our study, which includes the expert-annotated explanations dataset and the pseudonymized reader study data, are accessible on Figshare: https://figshare.com/s/c7feb070d066a4ccce19. Source data are provided with this paper.

## Code availability

All code used in this project is available on GitHub at https://github.com/tchanda90/Derma-XAI. This includes the source code for any custom software, scripts used in the analysis, as well as any data processing or visualisation code[63].

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

## Acknowledgements

We would like to thank Tim Holland-Letz for his statistical feedback. We gratefully acknowledge the collaborating clinicians listed in the Supplementary Data 1 who actively and voluntarily spent time to participate in the reader study. Some participants opted to not be mentioned despite their participation and the authors also thank these colleagues for their valuable contribution.

## Author contributions

T.J.B., T.C., K.H., S.Ho. and T.C.B. conceived of and designed the overall study. K.H. developed the explanatory ontology in accordance with S.Ho., R.L.B. and M.L.V. S.Ho., C.N.G., H.K., P.T., C.N., S.Po., E.C., I.C., J.M., L.A., T.F., S.P., S.S. and İ.Ö. annotated the dermoscopic images with explanations. K.H., S.Ho., G.P., S.K., F.F.G., M.H., M.E. and S.H. reviewed and improved the reader study design in the pilot study. T.J.B. and T.C.B. were responsible for explanation annotator recruitment. T.C. was responsible for data processing and management. T.C. and T.J.B. developed the explanatory classifier with dermatological expert feedback from S.Ho., T.J.B. and C.N.G. T.C. conducted experimental evaluation. T.C.B., T.J.B., S.Ho., K.D., M.G., B.S., J.S.U. and K.G. recruited reader study participants. T.C., K.H., T.J.B. and T.C.B. set up the web-based reader platforms. T.C., T.C.B. conducted the web-based reader study. T.C., C.W. and T.J.B. conducted statistical data analysis. T.C. generated the figures. W.S., S.F., E.K. and all other authors provided clinical and/or machine learning expertise and contributed to the interpretation of the results. T.C., K.H., S.Ho., T.C.B. and T.J.B. wrote the manuscript with input from all authors. All authors reviewed and corrected the final manuscript and collectively made the decision to submit for publication.

## Funding

Federal Ministry of Health, Berlin, Germany, grant number 2520DAT801, TJB. Ministry of Social Affairs, Health and Integration of the Federal State Baden-Württemberg, Germany, grant number 53 – 5400.1-007/5, TJB. Open Access funding enabled and organized by Projekt DEAL.

## Competing interests

PT reports grants from Lilly, consulting fees from Silverchair, lecture honoraria from Lilly, FotoFinder and Novartis, outside of the present publication. TJB owns a company that develops mobile apps (Smart Health Heidelberg GmbH, Heidelberg, Germany), outside of the scope of the submitted work. WS received travel support for participation in congresses and / or (speaker) honoraria as well as research grants from medi GmbH Bayreuth, Abbvie, Almirall, Amgen, Bristol-Myers Squibb, Celgene, GSK, Janssen, LEO Pharma, Lilly, MSD, Novartis, Pfizer, Roche, Sanofi Genzyme, and UCB outside of the present publication. MLV received travel support for participation in congresses and / or (speaker) honoraria as well as research grants from Abbvie, Almirall, Amgen, Bristol-Myers Squibb, Celgene, Janssen, Kyowa Kirin, LEO Pharma, Lilly, MSD, Novartis, Pfizer, Roche, Sanofi Genzyme, and UCB outside of the present publication. BS is on the advisory board or has received honoraria from Immunocore, Almirall, Pfizer, Sanofi, Novartis, Roche, BMS and MSD, research funding from Novartis and Pierre Fabre Pharmaceuticals, and travel support from Novartis, Roche, Bristol-Myers Squibb and Pierre Fabre Pharma, outside the submitted work. SH is on the advisory board or has received honoraria from Novartis, Pierre Fabre, BMS and MSD outside the submitted work. KD has received honoraria from Novartis, Pierre Fabre and Roche outside the submitted work. SF reports consulting or advisory board membership: Bayer, Illumina, Roche; honoraria: Amgen, Eli Lilly, PharmaMar, Roche; research funding: AstraZeneca, Pfizer, PharmaMar, Roche; travel or accommodation expenses: Amgen, Eli Lilly, Illumina, PharmaMar, Roche. JSU is on the advisory board or has received honoraria and travel support from Amgen, Bristol Myers Squibb, GSK, Immunocore, LeoPharma, Merck Sharp and Dohme, Novartis, Pierre Fabre, Roche, Sanofi outside the submitted work ME has received honoraria and travel expenses from Novartis and Immunocore. SHo received travel support for participation in congresses, (speaker) honoraria and research grants from Almirall, UCB, Janssen, Novartis, LEO Pharma and Lilly outside of the present publication. SP has received travel support for participation in congresses and/or speaker honoraria from Abbvie, Lilly, MSD, Novartis, Pfizer and Sanofi outside of the present publication. SPo is on the advisory board or has received honoraria from Galenicum Derma, ISDIN, Cantabria Labs and Mesoestetic. RLB has received support from Castle Bioscience for the International Melanoma Pathology Study Group Symposium and Workshop. MG served as consultant to argenx (honoraria paid to institution) and Almirall and received honoraria for participation in advisory boards / travel support from Biotest, GSK, Janssen, Leo Pharma, Lilly, Novartis and UCB - all outside the scope of the submitted work. MVH received honoraria from MSD, BMS, Roche, Novartis, Sun Pharma, Sanofi, Almirall, Biofrontera, Galderma. The other authors declare no competing interests.

## Additional information

[1]Digital Biomarkers for Oncology Group, German Cancer Research Center (DKFZ), Heidelberg, Germany. [2]Department of Dermatology, University Hospital, Technical University Dresden, Dresden, Germany. [3]Medical Faculty of University Heidelberg, Heidelberg, Germany. [4]Department of Dermatology, Medical University of Vienna, Vienna, Austria. [5]Department of Dermatology, Escuela de Medicina, Pontificia Universidad Católica de Chile, Santiago, Chile. [6]Dermatology Department, Hospital Clínic of Barcelona, University of Barcelona, IDIBAPS, Barcelona, Spain. [7]1st Department of Pathology, Medical School, National & Kapodistrian University of Athens, Athens, Greece. [8]Department of Dermatovenereology, Sestre milosrdnice University Hospital Center, Zagreb, Croatia. [9]Derma Style, Dermatovenerology clinic, Belgrade, Serbia. [10]Department of Dermatology, Dubai London Clinic, Dubai, United Arab Emirates. [11]West Dermatology, Newport Beach, California, USA. [12]Department of Dermatovenereology, Clinical Hospital Center Rijeka, Faculty of Medicine, University of Rijeka, Rijeka, Croatia. [13]LaserMed, Tallinn, Estonia. [14]Department of Dermatology, Faculty of Medicine, Gazi University, Ankara, Turkey. [15]Department of Translational Research, Institut Curie, Unit of Formation and Research of Medicine University of Paris, Paris, France. [16]Universidad Autónoma de Madrid, Madrid, Spain. [17]Charité - Universitätsmedizin Berlin, corporate member of Freie Universität Berlin and Humboldt-Universität zu Berlin, Department of Dermatology, Venereology and Allergology, Berlin, Germany. [18]Department of Dermatology, University Hospital Essen, University Duisburg-Essen, Essen, Germany. [19]Department of Dermatology, Uniklinikum Erlangen, Friedrich-Alexander-Universität Erlangen-Nürnberg, Erlangen, Germany. [20]Department of Dermatology, University Hospital Regensburg, Regensburg, Germany. [21]Department of Dermatology, Venereology and Allergology, University Hospital Würzburg, Würzburg, Germany. [22]Department of Dermatology, Venereology and Allergology, University Medical Center Mannheim, Ruprecht-Karl University of Heidelberg, Mannheim, Germany. [23]Division of Translational Medical Oncology, National Center for Tumor Diseases (NCT) Heidelberg and German Cancer Research Center (DKFZ), Heidelberg, Germany. [98]These authors contributed equally: Tirtha Chanda, Katja Hauser, Sarah Hobelsberger. *A list of authors and their affiliations appears at the end of the paper. ✉e-mail: titus.brinker@dkfz.de

## Reader Study Consortium

Alexander Salava[24], Alexander Thiem[25], Alexandris Dimitrios[26], Amr Mohammad Ammar[27], Ana Sanader Vučemilović[28], Andrea Miyuki Yoshimura[29], Andzelka Ilieva[30], Anja Gesierich[21], Antonia Reimer-Taschenbrecker[31,32], Antonios G. A. Kolios[33], Arturs Kalva[34], Arzu Ferhatosmanoğlu[35], Aude Beyens[36,37], Claudia Pföhler[38], Dilara Ilhan Erdil[39], Dobrila Jovanovic[40], Emoke Racz[41], Falk G. Bechara[42], Federico Vaccaro[43], Florentia Dimitriou[33], Gunel Rasulova[44], Hulya Cenk[45], Irem Yanatma[46], Isabel Kolm[33], Isabelle Hoorens[36], Iskra Petrovska Sheshova[47], Ivana Jocic[48], Jana Knuever[49], Janik Fleißner[21], Janis Raphael Thamm[50], Johan Dahlberg[51], Juan José Lluch-Galcerá[52], Juan Sebastián Andreani Figueroa[53], Julia Holzgruber[54], Julia Welzel[55], Katerina Damevska[56], Kristine Elisabeth Mayer[57], Lara Valeska Maul[58], Laura Garzona-Navas[59], Laura Isabell Bley[60], Laurenz Schmitt[61], Lena Reipen[21], Lidia Shafik[62], Lidija Petrovska[63], Linda Golle[64], Luise Jopen[65], Magda Gogilidze[66], Maria Rosa Burg[67], Martha Alejandra Morales-Sánchez[68], Martyna Sławińska[69], Miriam Mengoni[65], Miroslav Dragolov[70], Nicolás Iglesias-Pena[71], Nina Booken[67], Nkechi Anne Enechukwu[72], Oana-Diana Persa[57], Olumayowa Abimbola Oninla[73], Panagiota Theofilogiannakou[74], Paula Kage[75], Roque Rafael Oliveira Neto[76], Rosario Peralta[77], Rym Afiouni[78], Sandra Schuh[55], Saskia Schnabl-Scheu[79], Seçil Vural[80], Sharon Hudson[81], Sonia Rodriguez Saa[82], Sören Hartmann[83], Stefana Damevska[84], Stefanie Finck[85], Stephan Alexander Braun[86,87], Tim Hartmann[79], Tobias Welponer[88], Tomica Sotirovski[89], Vanda Bondare-Ansberga[90], Verena Ahlgrimm-Siess[88], Verena Gerlinde Frings[21], Viktor Simeonovski[89], Zorica Zafirovik[89], Julia-Tatjana Maul[31,32], Saskia Lehr[91], Marion Wobser[21], Dirk Debus[92], Hassan Riad[93], Manuel P. Pereira[17], Zsuzsanna Lengyel[94], Alise Balcere[95], Amalia Tsakiri[96] & Ralph P. Braun[97]

[24]Department of Dermatology and Allergology, Helsinki University Hospital, Helsinki, Finland. [25]Clinic and Policlinic for Dermatology and Venereology, University Medical Center Rostock, Rostock, Germany. [26]Department of Oncology, Evangelismos General Hospital of Athens, Athens, Greece. [27]Al-Azhar University, Cairo, Egypt. [28]Clinical Hospital Centre Split, Split, Croatia. [29]State University of Campinas, São Paulo, Brazil. [30]Private Clinical Hospital Zan Mitrev Clinic Skopje, Skopje, North Macedonia. [31]Department of Dermatology, Northwestern University, Feinberg School of Medicine, Chicago, IL, USA. [32]Medical Center – Department of Dermatology, University of Freiburg, Freiburg, Germany. [33]Department of Dermatology, University Hospital Zurich, Zurich, Switzerland. [34]Department of Public Health and Epidemiology, Riga Stradins University, Riga, Latvia. [35]Department of Dermatology and Venerology, Karadeniz Technical University Faculty of Medicine, Istanbul, Turkey. [36]Department of Dermatology, Ghent University Hospital, Ghent, Belgium. [37]Center for Medical Genetics Ghent, Ghent University Hospital, Ghent, Belgium. [38]Clinic for Dermatology, Venereology and Allergology, Saarland University Hospital, Homburg/Saar, Germany. [39]Istanbul Teaching and Research Hospital, Istanbul, Turkey. [40]Sloga Medik Center, Kruševac, Serbia. [41]Department of Dermatology, University Medical Center Groningen, University of Groningen, Groningen, the Netherlands. [42]Department of Dermatology, Venereology and Allergology, Ruhr-University Bochum, Bochum, Germany. [43]Università degli studi di Messina, Messina, Italy. [44]Koç University School of Medicine, Department of Dermatology and Venereology, Istanbul, Turkey. [45]Department of Dermatology, Pamukkale University, Denizli, Turkey. [46]Fethiye State Hospital, Fethiye, Turkey. [47]Dermatology Aesthetic Clinic Dr.Iskra, Skopje, North Macedonia. [48]Department of Dermatology and Venereology, School of Medicine, Military Medical Academy, Belgrade, Serbia. [49]Department of Dermatology, University Hospital Cologne, University of Cologne, Cologne, Germany. [50]Department of Dermatology and Allergology, University Hospital Augsburg, Augsburg, Germany. [51]Norrlands Universitetssjukhus, Umeå, Sweden. [52]Dermatology Department, Hospital Universitari Germans Trias i Pujol, Barcelona, Spain. [53]Dermatology Department, Universidad del Desarrollo, Facultad de Medicina Clínica Alemana, Santiago, Region Metropolitana, Chile. [54]Kepler University Hospital, Johannes Kepler University, Linz, Austria. [55]Clinic for Dermatology and Allergology, University Hospital Augsburg, Augsburg, Germany. [56]Faculty of Medicine, Ss Cyril and Methodius University in Skopje, Skopje, Macedonia. [57]Department of Dermatology and Allergology, Technical University of Munich, Munich, Germany. [58]Derpartment of Dermatology, University Hospital Basel, Basel, Switzerland. [59]Hospital Clinica Biblica, Costa Rica, Costa Rica. [60]Department of Dermatology, University Clinic Münster, Münster, Germany. [61]Department of Dermatology, University Hospital Heidelberg, Heidelberg, Germany. [62]East Surrey Hospital, Surrey and Sussex Healthcare Trust, Surrey, United Kingdom. [63]Dermatology department, PHI Clinical hospital Shtip, Shtip, Republic of North Macedonia. [64]Department of

Dermatology and Venereology, University Hospital Halle (Saale), Halle (Saale), Germany. [65]Department of Dermatology, University Hospital Magdeburg, Magdeburg, Germany. [66]International Dermatological Clinic Kani, Kvemo Kartli, Georgia. [67]Department of Dermatology and Venereology, University Medical Center Hamburg- Eppendorf | UKE, Hamburg, Germany. [68]Centro Dermatológico Dr. Ladislao de la Pascua, Ciudad de Mexico, Mexico. [69]Department of Dermatology, Venereology and Allergology, Faculty of Medicine, Medical University of Gdańsk, Gdańsk, Poland. [70]Mirabel Clinic, Burgas, Bulgaria. [71]Dermatology Department, IMQ San Rafael, A Coruña, Spain. [72]Department of Medicine, Nnamdi Azikiwe University, Nnewi, Anambra, Nigeria. [73]Department of Dermatology and Venereology, Faculty of Clinical Sciences, College of Health Sciences, Obafemi Awolowo University, Ile-Ife, Osun State, Nigeria. [74]Dermatology Department, Evangelismos General Hospital of Athens, Athens, Greece. [75]Hautarztpraxis Dr. med. Paula Kage, Freiberg, Germany. [76]Federal University of Mato Grosso UFMT, Mato Grosso, Brazil. [77]Medical Research Institute A. Lanari, University of Buenos Aires, Buenos Aires, Argentina. [78]Faculty of Medicine, Saint-Joseph University, Beirut, Lebanon. [79]Department of Dermatology, University Hospital Tübingen, Tübingen, Germany. [80]Department of Dermatology, Koç University School of Medicine, Istanbul, Turkey. [81]Clinical Nurse Consultant, Melbourne, Australia. [82]Dermatology Department, Hospital El Carmen, Mendoza, Argentina. [83]Department of Dermatology, University of Dresden, Dresden, Germany. [84]Department of Dermarology and Venereology, Acibadem CityClinic Tokuda Hospital, Sofia, Bulgaria. [85]Centre for Dermatology, Venereology and Allergology, Stuttgart Hospital, Stuttgart, Germany. [86]Department of Dermatology, University Hospital Muenster, Muenster, Germany. [87]Department of Dermatology, Medical Faculty, Heinrich-Heine University, Duesseldorf, Germany. [88]Department of Dermatology and Allergology, Paracelsus Medical University Salzburg, Salzburg, Austria. [89]University Clinic of Dermatology, Faculty of Medicine, University "Ss. Cyril and Methodius", Skopje, North Macedonia. [90]Rîga 1st Hospital, Riga, Latvia. [91]Department of Dermatology and Venereology, Medical Center-University of Freiburg, Faculty of Medicine, University of Freiburg, Freiburg, Germany. [92]Department of Dermatology, Nuremberg General Hospital, Paracelsus Medical University, Nuremberg, Germany. [93]HMC, Doha Metropolitan Area, Doha, Qatar. [94]Department of Dermatology, Venerology and Oncodermatology, University of Pécs, Pécs, Hungary. [95]Riga Stradins University Department of Dermatology and Venereology, Riga, Latvia. [96]Private Clinic, Thessaloniki, Greece. [97]Department of Dermatology, University Hospital Zürich and University of Zürich, Zürich, Switzerland.

