## [Peer Review File · Nature Communications]

REVIEWER COMMENTS

Reviewer #1 (Remarks to the Author): expertise in melanoma pathology

Dear Authors,

your work intends to close the interpretability gap in AI-based decision support systems by developing an XAI that can produce domain-specific interpretable explanations to aid in melanoma diagnosis. Additionally, you evaluate the XAI system's effect on clinicians' diagnostic accuracy, confidence, and trust in the system, and assessed the factors that contribute to trust. A multi-modal XAI system with clinical explanations was created and conducted the first large-scale reader study on such an XAI system in dermatology.

The results showed that the diagnostic performance of our XAI was on par with the baseline, while being interpretable by learning human-relevant features. Additionally, XAI was minimally affected by spurious correlations in two ways.

Will the work be of significance to the field and related fields? How does it compare to the established literature? The study is original although limited by the "artificial" conditions but the Authors stated it in the discussion. Methodology and data analysis are well done and conclusions are strongly supported by data.

There are enough detail provided in the methods for the work to be reproduced by other scientists.

Reviewer #2 (Remarks to the Author): expertise in machine learning in pathology and neural networks

In this paper, the authors have developed an explainable artificial intelligence (XAI) system based on two existing classifiers, which can produce text- and region-based explanations that are easily interpretable by dermatologists alongside its differential diagnoses of melanomas and nevi. This work promotes the development of human-machine collaboration in medicine by developing a multi-modal XAI to support its decision-making and evaluate its impact on clinical doctors. The experimental validation has shown that the proposed XAI achieves good diagnostic accuracy and the results are strongly aligned with the clinician's interpretation. Although this work is of some research implications for the development of interpretable machine learning in the field of computer-aided diagnosis, the following issues that need to be further addressed by the authors.

1. It is recommended that the authors analyze the innovation of the proposed XAI and conventional interpretable AI methods from the perspective of machine learning algorithm principles.

2. How to define and quantify the mentioned transparency while applying AI methods.

3. In terms of classifier design, it seems that the conventional ResNet algorithm is used, and where is its innovation mainly reflected?

4. For computer-aided diagnostic algorithms implemented using AI methods, it is recommended that the authors provide an analytical review of relevant advanced data processing and model design work in terms of attention mechanism in XAI, such as the work [a1] for reference. [a1] AGGN: Attention-based glioma grading network with multi-scale feature extraction and multi-modal information fusion, Computers in Biology and Medicine.

5. How about the generalization ability of the proposed XAI in other computer-aided-diagnosis applications.

6. Since it is proven that the trust is related to the overlap between human and machine explanation, then in this regard, will it be possible to induce the attention of AI towards the preference of human.

7. What is the basis for determining the determinacy of the classifier by setting the threshold of the temperature scale to 0.7?

8. What are the benefits of interpretable methods based on polygons and heatmaps, respectively?

9. What are the motivation and benefits of machine learning approaches that fuse multi-modal data in the designed interpretation scheme, including visual and text-based components and the evaluation of classification confidence?

10. In the experimental setup, the proposed XAI is only compared with the baseline method, while a comprehensive analysis with other state-of-the-art algorithms is missing. Moreover, enough details should be provided in the methods for the work to be reproduced.

RESPONSE TO REVIEWERS' COMMENTS

1.It is recommended that the authors analyze the innovation of the proposed XAI and conventional interpretable AI methods from the perspective of machine learning algorithm principles.

Thank you for your comment. We have extended the machine learning literature presented in the Introduction section by a summary of the state-of-the-art of XAI in digital dermatology (Introduction section, yellow text), a broader overview over XAI in general as relevant for the application in diagnosis assistance systems and the closing of the interpretability gap (Introduction section, blue text) as well as a discussion of the relation of commonly used XAI algorithms to the points raised by Rudin (Introduction section, yellow text). Additionally, an extension on the work by Jalaboi et al. and Lucieri et al. whose work is closest to ours has been added (Introduction section, yellow text).

However, the main innovation of our XAI is not so much technical innovation, but domain-specific improvement. In contrast to previous work by Jalaboi et al., Tschandl et al., and Lucieri et al. we

- Close the interpretability gap in a way that is tailored to dermatologists as users
- Move our XAI beyond the proof-of-concept stage of Lucieri et al. and Jalaboi et al. by using a sufficiently large expert annotated dataset (roughly 3600 images in our case, compared to roughly 1000 (Lucieri et al.) and roughly 550 images (Jalaboi et al.)).
- Successfully evaluate the influence of our XAI on dermatologists (in contrast to Tschandl et al. who found that their XAI system was ignored by readers)
- Achieve state-of-the-art performance as measured by the balanced accuracy

2.How to define and quantify the mentioned transparency while applying AI methods.

Thank you for raising this point. To address this, we performed additional experiments in an attempt to quantify the transparency. We added a paragraph to the Results section (subsection "Our XAI achieves good diagnostic accuracy", green text) of the manuscript.

3. In terms of classifier design, it seems that the conventional ResNet algorithm is used, and where is its innovation mainly reflected?

Thank you for bringing this issue to our attention. We do use a ResNet50 as the backbone for our XAI framework. This particular ResNet (here: both architecture and training details) has been chosen for two main reasons. First, the network won the lesion classification task of the ISIC 2017 challenge (classification of melanoma vs. nevus/7 classes)^{1,2}. Therefore, the network's classification performance has been demonstrated in a reliable and reproducible way.

Second, to promote reproduction of our work, we chose the ResNet50 architecture as it is accessible in terms of requirements to computational power for training (compare to the ablations done for comment 10/ Extended Data Table 1) and presence in common, well-maintained deep learning libraries such as PyTorch³.

The innovation of our paper is mainly reflected in the design of the XAI, i.e. that by design it closes the interpretability gap described by Rudin⁴ and thus can address concerns about XAI safety in high-stakes situations. Specifically, our approach closes the interpretability gap by using the expert annotations (both explanatory image regions and highly domain-specific text explanations) for training such that the XAI produces text- and region-based explanations.

While ours is not the most innovative study in terms of neural network design, we believe that the novel XAI design evaluated in a large-scale reader study closes a gap in the current research literature on XAI in dermatology.

We added paragraphs to the Introduction section (green) that address this issue.

4. For computer-aided diagnostic algorithms implemented using AI methods, it is recommended that the authors provide an analytical review of relevant advanced data processing and model design work in terms of attention mechanism in XAI, such as the work [a1] for reference.[a1] AGGN: Attention-based glioma grading network with multi-scale feature extraction and multi-modal information fusion, Computers in Biology and Medicine.

Thank you for bringing this issue to our attention. We added several paragraphs in the Introduction part of the paper (highlighted in yellow) to give a better overview over the state of XAI in dermatology as well as the most relevant related work.

5.How about the generalization ability of the proposed XAI in other computer-aided-diagnosis applications.

Thank you for the question! Our XAI is easily generalizable to other computer-aided-diagnosis tasks, as long as a suitable descriptive framework is present in the target domain. By suitable we mean a framework that has (1) been established by medical experts in the domain, is (2) standardised and (3) part of the diagnostic process so that clinicians in the field are familiar with it. An example for such a descriptive framework can be found in Gleason Scoring^{5,6} in uropathology.

Otherwise, we expect that clinicians familiar with the given descriptive framework will benefit more from our XAI than those unfamiliar. We reported this as a finding in the Results Section (AI support improves diagnostic accuracy, but XAI support does not further increase diagnostic accuracy over AI support alone): we found that dermatologists familiar with the descriptive

terminology coined by Kittler et al.⁷ benefit more from our XAI than clinicians unfamiliar with the terminology.

We have added a paragraph in the Discussion section (green) that addresses these points.

6. Since it is proven that the trust is related to the overlap between human and machine explanation, then in this regard, will it be possible to induce the attention of AI towards the preference of human.

Thank you for addressing this point. Our XAI is by design tailored towards human preference: First, it is designed to use an ontology that encapsulates state-of-the-art medical knowledge on melanoma diagnosis. Second, for its training expert-annotated data, i.e. localisations of known dermoscopic features of melanomas and nevi, is used to tailor our XAI's explanations to the human understanding of the task.

We have added paragraphs clarifying and emphasising this feature of our XAI to the Introduction (cyan text) and the Methods section (under "XAI Development, Classifier design", green text) of the manuscript.

7. What is the basis for determining the determinacy of the classifier by setting the threshold of the temperature scale to 0.7?

Thanks for bringing up this issue. We believe that 0.7 as a calibrated confidence threshold is a reasonable rule of thumb. Prior work in our group determined the optimal threshold between certain and uncertain predictions. Experimenting with various thresholds by rejecting uncertain predictions, our colleagues discovered that a cutoff of 0.7 improved accuracy the most⁸. It should be noted, however, that they experimented with uncalibrated uncertainty scores, whereas we use calibrated scores based on temperature scaling. We have added text to clarify this in the Methods section (subsection "Design of the explanations subsection", blue text).

8. What are the benefits of interpretable methods based on polygons and heatmaps, respectively?

Thank you for raising this question. Interpretation methods that can be visualised with heatmaps or polygons are designed to reveal the saliency information of an image in a pixel-wise manner - i.e. to reveal which image regions are important for a network decision. The choice between a polygon and a heatmap is thus not a question of the method per se, but a question of the visualisation needs. Heatmaps typically visualise the pixel-wise importance by mapping the saliency values to a colormap and overlaying the respective values directly over the input image in an opaque or semi-transparent manner. The benefit of this approach is that fine-grained saliency information is communicated to the observer. This can be especially helpful to debug or

finetune a network. However, details of the image itself are occluded when using a heatmap. Polygons, on the other hand, only indicate a saliency threshold and can thus be overlaid over the input image with minimal occlusion. This is clearly desirable when the visibility of the input image is paramount to the task, as was the case in our reader study.

We have added two additional sentences to the Methods section (subsection “Design of the explanations”, green text) highlighting the benefits of heatmap visualisations and clarifying the drawbacks of polygon methods regarding the loss of fine-grained saliency information. We deem that the benefits of polygon visualisations for our case (unobstructed view of the skin lesion as requested by our consulting dermatologists) is sufficiently motivated in the text as is.

9. What are the motivation and benefits of machine learning approaches that fuse multi-modal data in the designed interpretation scheme, including visual and text-based components and the evaluation of classification confidence?

Thank you for asking this important question. While designing the interpretation scheme, we first considered working with unimodal explanations on the image by highlighting regions of interest. However, as pointed out in the Introduction section, this unimodal design does not close the interpretability gap defined by Rudin.

Miller et al⁹. who integrates scientific literature from social sciences for recommending XAI design argues that explanations are always contextual. Although events may have multiple causes, the explainee typically concerns themselves only with a small subset that is relevant to the given context. Motivated by our objective of user-centricity, we aimed to design the XAI explanations in the manner in which a dermatologist would explain their diagnostic decision. Exchange with our consulting dermatologists revealed their preference for textual display of the patterns alongside localization within the image and an indication of AI classification confidence.

Furthermore, results of a survey conducted by our DKFZ research group (currently under review), which evaluated the criteria prioritised by dermatologists for diagnostic AI support systems, confirmed that the majority of respondents favoured the combination of image and text explanations over other suggested unimodal and multimodal options.

Thus, we arrived at the conclusion that our multimodal explanation scheme served our task-specific requirements for AI explanations the best.

Regarding the broader advantages of multimodal explanation schemes compared to unimodal designs, unfortunately, the general computer vision literature on XAI lacks studies that investigate the superiority of either approach.

The effect of multimodal explanations and their ablations for diagnostic AI support systems in medicine is an open question that has captured our attention during this work. Accordingly, we are presently devising plans to explore and address this issue in our forthcoming research. We have added this information to the Discussion section (orange text).

10. In the experimental setup, the proposed XAI is only compared with the baseline method, while a comprehensive analysis with other state-of-the-art algorithms is missing. Moreover, enough details should be provided in the methods for the work to be reproduced.

Thank you for your valuable input. We agree that the manuscript would benefit from a comprehensive analysis with multiple CNN architectures as well as comparisons with other state-of-the-art (SOTA) approaches. To this end, we performed additional experiments with eight CNN architectures and added the necessary text in the Results section (subsections “Our XAI achieves good diagnostic accuracy”, blue text; and subsection “Our XAI is strongly aligned with clinicians’ explanations, blue text) and Discussion section (blue text) of the manuscript.

Additionally, we reproduced two SOTA approaches for skin cancer classification. The first approach is based on attention mechanisms and reports competitive performance on the ISIC 2017 dataset^{1,10}. We chose this approach as it provides explainability through saliency maps while achieving competitive classification performance. The second approach is an ensemble method that achieved the winning position on the ISIC 2020 challenge¹¹. This approach achieves SOTA classification performance, but lacks interpretability since it’s an ensemble. We’ve added text explaining this in the Results (subsection “Our XAI achieves good diagnostic accuracy”, blue text) and Discussion (blue text). Additionally, we’ve moved some text from Supplementary Information E to the Methods (subsection “XAI Development”, yellow text) necessary to reproduce the work.

Literature

1. Codella, N. C. F. *et al.* Skin lesion analysis toward melanoma detection: A challenge at the 2017 International symposium on biomedical imaging (ISBI), hosted by the international skin imaging collaboration (ISIC). in *2018 IEEE 15th International Symposium on Biomedical Imaging (ISBI 2018)* 168–172 (2018). doi:10.1109/ISBI.2018.8363547.
2. Matsunaga, K., Hamada, A., Minagawa, A. & Koga, H. Image Classification of Melanoma, Nevus and Seborrheic Keratosis by Deep Neural Network Ensemble. Preprint at

<http://arxiv.org/abs/1703.03108> (2017).

3. Paszke, A. *et al.* PyTorch: An Imperative Style, High-Performance Deep Learning Library. in *Advances in Neural Information Processing Systems* vol. 32 (Curran Associates, Inc., 2019).
4. Rudin, C. Stop explaining black box machine learning models for high stakes decisions and use interpretable models instead. *Nat. Mach. Intell.* **1**, 206–215 (2019).
5. van Leenders, G. J. L. H. *et al.* The 2019 International Society of Urological Pathology (ISUP) Consensus Conference on Grading of Prostatic Carcinoma. *Am. J. Surg. Pathol.* **44**, e87–e99 (2020).
6. Tateo, V., Mollica, V., Rizzo, A., Santoni, M. & Massari, F. Re: WHO Classification of Tumours, 5th Edition, Volume 8: Urinary and Male Genital Tumours. *Eur. Urol.* S0302-2838(23)02792–6 (2023) doi:10.1016/j.eururo.2023.04.030.
7. Kittler, H. & Tschandl, P. *Dermatoskopie: Musteranalyse pigmentierter und unpigmentierter Hautläsionen*. (Facultas, 2015).
8. Höhn, J. *et al.* Combining CNN-based histologic whole slide image analysis and patient data to improve skin cancer classification. *Eur. J. Cancer* **149**, 94–101 (2021).
9. Miller, T. Explanation in artificial intelligence: Insights from the social sciences. *Artif. Intell.* **267**, 1–38 (2019).
10. Datta, S. K., Shaikh, M. A., Srihari, S. N. & Gao, M. Soft Attention Improves Skin Cancer Classification Performance. in *Interpretability of Machine Intelligence in Medical Image Computing, and Topological Data Analysis and Its Applications for Medical Data* (eds. Reyes, M. *et al.*) 13–23 (Springer International Publishing, 2021). doi:10.1007/978-3-030-87444-5_2.
11. Ha, Q., Liu, B. & Liu, F. Identifying Melanoma Images using EfficientNet Ensemble:

Winning Solution to the SIIM-ISIC Melanoma Classification Challenge. (2020)

doi:10.48550/arxiv.2010.05351.

REVIEWERS' COMMENTS

Reviewer #2 (Remarks to the Author):

In the revised manuscript, the authors have addressed my previous concerns. Thus, I recommend the acceptance in the journal.